# *Memorize What Matters*:
# Emergent Scene Decomposition from Multitraverse

**Yiming Li**[1,2]   **Zehong Wang**[1]   **Yue Wang**[2,3]   **Zhiding Yu**[2]
**Zan Gojcic**[2]   **Marco Pavone**[2,4]   **Chen Feng**[1]   **Jose M. Alvarez**[2]
[1]NYU   [2]NVIDIA   [3]USC   [4]Stanford University

Project Page: https://nvlabs.github.io/3DGM/

## Abstract

Humans naturally retain memories of permanent elements, while ephemeral moments often slip through the cracks of memory. This selective retention is crucial for robotic perception, localization, and mapping. To endow robots with this capability, we introduce 3D Gaussian Mapping (3DGM), a *self-supervised*, *camera-only* offline mapping framework grounded in 3D Gaussian Splatting. 3DGM converts multitraverse RGB videos from the same region into a Gaussian-based environmental map while concurrently performing 2D ephemeral object segmentation. Our key observation is that the environment remains consistent across traversals, while objects frequently change. This allows us to exploit self-supervision from repeated traversals to achieve environment-object decomposition. More specifically, 3DGM formulates multitraverse environmental mapping as a robust 3D representation learning problem, treating pixels of the environment and objects as inliers and outliers, respectively. Using robust feature distillation, feature residual mining, and robust optimization, 3DGM simultaneously performs 2D segmentation and 3D mapping without human intervention. We build the Mapverse benchmark, sourced from the Ithaca365 and nuPlan datasets, to evaluate our method in unsupervised 2D segmentation, 3D reconstruction, and neural rendering. Extensive results verify the effectiveness and potential of our method for self-driving and robotics.

## 1   Introduction

Vision-based 3D mapping is essential for autonomous driving but faces two key challenges: *(1)* dynamic objects disrupting multi-view consistency and *(2)* accurately reconstructing 3D structures from 2D images. Existing methods rely on pretrained segmentation models to filter out dynamic objects and LiDARs to enhance geometry. However, these approaches are limited by the need for human annotations during pretraining, along with the high costs and limited portability of LiDARs.

Motivated by the aforementioned challenges, we aim to develop a *self-supervised* and *camera-only* 3D mapping approach, reducing the reliance on human annotations and LiDARs. We consider a practical *multitraverse* driving scenario, where autonomous vehicles repeatedly traverse the same routes or regions at different times. During each traversal, the ego-vehicle encounters new pedestrians and vehicles, much like how humans navigate the same 3D environment but encounter different groups of passersby each day. Inspired by humans' ability to memorize the **permanent** and ignore the **ephemeral**[1] during repeated spatial navigation, we pose the following question:

---

[1]We will use *ephemeral* and *transient* interchangeably to refer to objects that temporarily appear or disappear across various traversals of the same location, such as pedestrians, vehicles, or other temporary elements.

38th Conference on Neural Information Processing Systems (NeurIPS 2024).

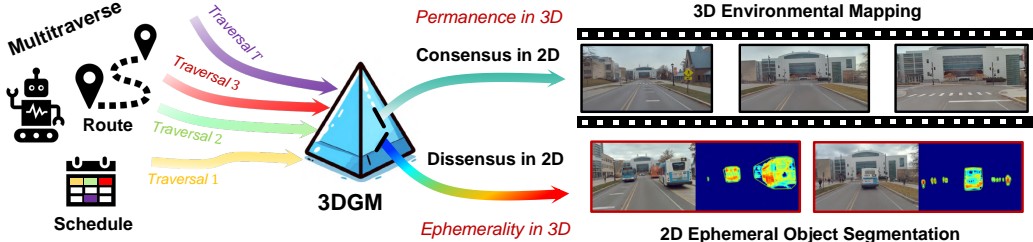

Figure 1: **A high-level diagram of** `3D Gaussian Mapping (3DGM)`**.** Given multitraverse RGB videos, 3DGM outputs a Gaussian-based environment map (`EnvGS`) and 2D ephemerality segmentation (`EmerSeg`). Note that the proposed framework is **LiDAR-free** and **self-supervised**.

*Can we develop an autonomous mapping system that identifies and memorizes only the consistent environmental structures in a 3D world across multiple traversals, without relying on human supervision?*

We provide an affirmative answer to this question. Our key insight lies in using the consensus across repeated traversals as a self-supervision signal, ensuring that the learned map retains only consensus structures (*permanent environment*) while discarding dissensus elements (*transient objects*). We ground this insight in 3D Gaussian Splatting (3DGS) [1], which models a 3D scene using a group of 3D Gaussians with learnable attributes such as position, color, and opacity. This scene representation offers both geometric and photometric information, benefiting various downstream applications in autonomous driving. We leverage abundant images from multiple traversals to facilitate Gaussian initialization using Structure from Motion (SfM) [2], without relying on LiDARs. Subsequently, we learn the environmental Gaussians from multitraverse RGB videos by minimizing the rendering loss.

To optimize a time-invariant 3D representation from input images containing time-varying structures, we frame *multitraverse environmental mapping* as a *robust representation learning* problem, where pixels from transient objects are treated as outliers. Specifically, we distill self-supervised robust features—denoised DINOv2 [3,4]—into Gaussians to facilitate outlier identification. We then employ a novel feature residual mining strategy to fully exploit the spatial information within the rendering loss map. This strategy aids in precise outlier grouping, improving transient object segmentation. Finally, we apply a robust loss function to optimize the 3D environmental Gaussians. As a result, we accurately learn the Gaussian-based environment map from inlier pixels and even generate 2D masks of transient objects for free, as illustrated in Fig. 1.

We build the *Mapping and segmentation through multitraverse* (**Mapverse**) benchmark, sourced from the Ithaca365 [5] and nuPlan [6] datasets to evaluate our method in three tasks: unsupervised 2D segmentation, 3D reconstruction, and neural rendering. Quantitative and qualitative results demonstrate the effectiveness of our method in autonomous driving scenarios.

To summarize, our key innovations are listed as follows.

- **Problem formulation**   We address the multitraverse RGB mapping problem through robust representation learning, treating pixels of the environment as inliers and objects as outliers.
- **Technical design**   We introduce feature residual mining to leverage spatial information from rendering loss maps, enabling more accurate outlier segmentation in self-driving scenes.
- **System integration**   We build 3D Gaussian Mapping (`3DGM`) that jointly generates 3D environmental Gaussians and 2D ephemerality masks without LiDARs and human annotations.
- **Dataset curation**   We build a large-scale multitraverse driving benchmark from real-world driving data, featuring **40** locations, each with no less than **10** traversals, totaling **467** driving video clips and **35,304** images. Code and data are released at `https://github.com/NVlabs/3DGM`.

## 2   Related Works

**Multitraverse driving**   A vehicle generally operates within the same geographical area, resulting in multiple traversals of the same location. This repetition enriches the vehicle's memory of specific

places, enhancing its capabilities in perception and localization [7–10]. Regarding perception, the Hindsight framework [11] utilizes past LiDAR point clouds to learn memory features that are easy to query, thereby addressing the challenges of point sparsity and boosting 3D detection performance. Other studies have employed the persistence prior score [12, 13], which quantifies the consistency of a single LiDAR point across multiple traversals, for self-training of detectors and domain adaptation. In localization, a significant number of works focus on either metric [14, 15] or topological [16, 17] localization, aiming to match a query image with a set of reference images collected from different traversals under varying seasonal or lighting conditions. Closely related to our work is [18], which employs multiple traversals to map out ephemeral regions, enhancing monocular visual odometry in dense traffic conditions. However, this approach also depends on the consistency of LiDAR point clouds across traversals, remarking an unexplored gap in leveraging consensus in the 2D image space.

**NeRF and 3DGS**    NeRF has recently revolutionized novel-view synthesis and scene reconstruction with image or video input, boasting a wide range of applications in graphics, vision, and robotics. NeRF employs a volumetric representation and trains neural networks to model density and color. The success of NeRF has sparked a surge in follow-up methods aiming to enhance quality [19–21] and increase speed [22–24]. The recent 3D Gaussian Splatting (3DGS) [1] uses an explicit Gaussian-based representation and splatting-based rasterization [25] to project anisotropic 3D Gaussians onto a 2D screen. It determines the pixel's color by performing depth sorting and $\alpha$-blending on the projected 2D Gaussians, thus avoiding the complex sampling strategy of ray marching and achieving real-time rendering. Subsequent works have applied 3DGS to scene editing [26], dynamic scene modeling [27, 28], sparse view reconstruction [29], mesh reconstruction [30], semantic understanding [31, 32], and indoor SLAM [33].

**NeRF and 3DGS for self-driving**    Beyond their use in object-centric scenarios and bounded indoor environments, NeRF and 3DGS have also been explored in unbounded driving scenes [34, 35]. Several works address the implicit surface reconstruction of static scenes [36–38]. A large body of research focuses on dynamic scene reconstruction from a single driving log. Most works use a compositional method and rely on bounding annotations/trained detectors to model dynamic objects [39–45]. EmerNeRF [46] is the first self-supervised method to learn 4D neural representations of driving scenes from LiDAR-camera recordings. It couples static, dynamic, and flow fields [24] and leverages the flow field to aggregate multi-frame information to enhance the feature representation of dynamic objects. Another line of research investigates the scalability of the neural representation to model large-scale scenes [47–52]. Block-NeRF [47] segments the scene into separately trained NeRF models, processing camera images from multiple drives, and applies a semantic segmentation model [53] to exclude common movable objects. SUDS takes the input of multitraverse driving logs, leveraging RGB images, LiDAR point clouds, DINO [54], and 2D optical flow [55] for dynamic scene decomposition. In this work, we create an environment map represented by 3DGS without requiring LiDARs, leveraging the multitraverse consensus for self-supervised object removal.

**Scene decomposition**    Traditional background subtraction approaches [56, 57] distinguish moving objects from static scenes by comparing successive video frames and identifying significant differences as foreground elements. Representative works include low-rank decomposition, which treats moving objects in the scene as pixel-wise sparse outliers [58, 59]. These methods are typically used in surveillance applications and are limited to static cameras. Follow-up works [60, 61] investigate background subtraction for mobile robotics, yet suffering from low performance. NeRF has recently emerged as a popular scene representation and has been applied to the self-supervised dynamic-static decomposition of indoor scenes by modeling *time-varying* and *time-independent* components separately [62, 63]. EmerNeRF [46] extends similar intuition to autonomous driving and obtains scene flow for free while achieving dynamic-static decomposition of a single traversal. Yet it still depends on the LiDAR inputs. In this study, we leverage signals of consensus and dissensus across multiple traversals to accomplish *permanence-ephemerality* decomposition using only image inputs.

**Vision foundation models**    Inspired by the success of scaling in NLP [64], the field of computer vision intensively studies large-scale self-supervised pre-training with Transformers [65]. Vision Transformers (ViTs) [66], pre-trained on extensive datasets, achieve excellent image recognition results. DINO [54] further amplifies feature representation capabilities by harnessing self-supervised learning alongside knowledge distillation. Meanwhile, scene layouts emerge within the attention maps, enabling unsupervised semantic understanding. DINOv2 [4] scales up both the data and model

size, achieving more robust visual features. Subsequent research focuses on examining noise artifacts to further enhance the performance of self-supervised descriptors, including training-free denoising of ViTs [3] and retraining ViTs with registered tokens [67]. In this work, we leverage denoised DINOv2 features [3,4] to facilitate consensus verification across multiple traversals in pixel space, *as the high-dimensional features prove more resilient to changes in environmental appearance.*

# 3 3DGS: 3D Gaussian Splatting

3D Gaussian Splatting [1] represents the 3D environment with a set of anisotropic 3D Gaussians, denoted by $\mathbf{G} = \{\mathbf{G}_i \mid i = 1, \ldots, N\}$, where $N$ is the total number of Gaussians. Each Gaussian, $\mathbf{G}_i$, is parameterized by its mean vector $\boldsymbol{\mu}_i \in \mathbb{R}^3$, indicating the position, and a covariance matrix $\boldsymbol{\Sigma}_i \in \mathbb{R}^{3\times3}$, defining its shape. To guarantee positive semi-definiteness, the covariance matrix $\boldsymbol{\Sigma}_i$ is further decomposed as $\boldsymbol{\Sigma}_i = \mathbf{R}_i \mathbf{S}_i \mathbf{R}_i^\top$, with $\mathbf{R}_i$ being an orthogonal rotation matrix and $\mathbf{S}_i$ a diagonal scaling matrix. These are stored compactly as a rotation quaternion $\mathbf{q}_i \in \mathbb{R}^4$ and a scaling factor $\mathbf{s}_i \in \mathbb{R}^3$. Each Gaussian also incorporates an opacity value $\alpha_i \in \mathbb{R}$ and a spherical harmonics coefficients $\boldsymbol{\beta}_i$. Therefore, the learnable parameters for the $i$-th Gaussian are $\mathbf{G}_i = [\boldsymbol{\mu}_i, \mathbf{q}_i, \mathbf{s}_i, \alpha_i, \boldsymbol{\beta}_i]$. Rendering from a viewpoint computes the color at pixel $\mathbf{p}$ (denoted by $\mathbf{c}_\mathbf{p}$) via volumetric rendering, integrating $K$ ordered Gaussians $\{\mathbf{G}_k \mid k = 1, \ldots, K\}$ overlapping pixel $\mathbf{p}$, *i.e.*, $\mathbf{c}_\mathbf{p} = \sum_{k=1}^{K} \mathbf{c}_k \alpha_k \prod_{j=1}^{k-1}(1 - \alpha_j)$. Here, $\alpha_k$ is derived by evaluating a 2D Gaussian projection [25] from $\mathbf{G}_k$ onto pixel $\mathbf{p}$, multiplied by the Gaussian's learned opacity, and $\mathbf{c}_k$ is the color obtained by evaluating the spherical harmonics of $\mathbf{G}_k$. The Gaussians are sorted by their depth from the viewpoint. The overall objective is to minimize the rendering loss:

$$\mathcal{L} = \sum_t \mathcal{L}_{rgb}(\mathbf{I}_t(\boldsymbol{\xi}_t; \mathbf{G}), \mathbf{I}_t) \tag{1}$$

where $\mathbf{I}_t(\boldsymbol{\xi}_t; \mathbf{G}) \in \mathbb{R}^{w\times h\times 3}$ is the RGB image indexed by $t$, with spatial dimensions $w \times h$ and rendered from the pose $\boldsymbol{\xi}_t \in \mathfrak{se}(3)$, given Gaussians $\mathbf{G}$. $\mathbf{I}_t \in \mathbb{R}^{w\times h\times 3}$ is the paired ground truth image. $\mathcal{L}_{rgb}$ is a loss function such as L1 loss. Initialized by COLMAP [2], all attributes of $\mathbf{G}$ are learned by executing this view reconstruction task. Meanwhile, adaptive densification and pruning strategies are proposed to improve the fitting of the 3D scene.

# 4 3DGM: 3D Gaussian Mapping

## 4.1 Problem Formulation

**Assumption** We make reasonable assumptions about the stability of the environment and the transience of objects within it. Specifically, we assume that there are no major environmental changes, a realistic expectation when data is collected over a certain period under consistent weather and lighting conditions. Meanwhile, we consider all movable objects to be transient; despite their potential static nature during a particular traversal, they are expected to eventually move somewhere else, allowing the camera to capture dissensus over time.

**Setup** We conduct offline mapping of a specified spatial area by repeatedly traversing it with vehicles equipped with a monocular camera. The 3D environment map is represented by a set of 3D Gaussians, denoted as $\mathbf{G} = \{\mathbf{G}_i \mid i = 1, \ldots, N\}$. Each $\mathbf{G}_i$ has a set of learnable parameters $[\boldsymbol{\mu}_i, \mathbf{q}_i, \mathbf{s}_i, \alpha_i, \boldsymbol{\beta}_i, \mathbf{f}_i]$, where $\mathbf{f}_i \in \mathbb{R}^d$ is a self-supervised $d$-dimensional semantic feature such as DINO [4] for a more robust representation, and other parameters follow 3DGS as detailed in Sec. 3. This mapping approach not only captures the geometry but also the photometry of the environment, yielding a comprehensive scene representation for downstream tasks such as geometry reconstruction and view synthesis. The input to our approach comes from a set of unposed images, sourced from multitraverse RGB videos, denoted by $\mathbf{I} = \{\mathbf{I}_t \in \mathbb{R}^{w\times h\times 3} \mid t = 1, \ldots, T\}$, where $T$ is the total number of images, $w$ and $h$ are the width and height of each image, respectively.

**Target** The target is to refine $\mathbf{G}$ to a level where it can accurately render images $\mathbf{I}_t(\boldsymbol{\xi}_t; \mathbf{G})$ that closely match the real images $\mathbf{I}_t$, captured from specific poses $\boldsymbol{\xi}_t$. Although $\mathbf{G}$ represents a 3D spatial map, the input images encompass 4D information with both spatial and temporal dimensions. Hence, our method needs to differentiate between the environment and ephemeral objects, like pedestrians

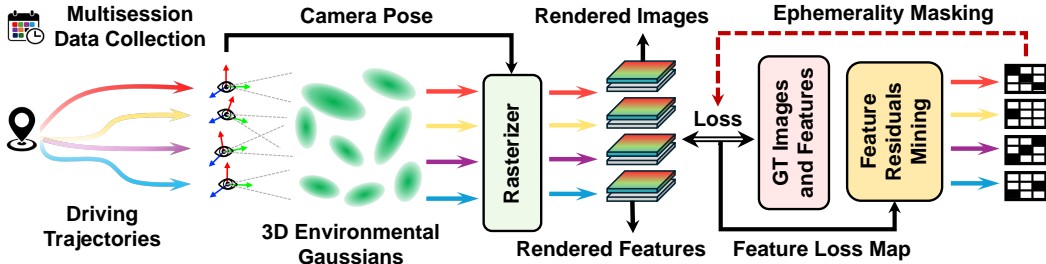

Figure 2: **An overall illustration of** 3DGM**.** Given RGB camera observations collected at different times, we use COLMAP to obtain the camera poses and initial Gaussian points. Then we utilize splatting-based rasterization to render both RGB images and robust features from the environmental Gaussians. We further leverage feature residuals to extract the object masks by mining spatial information of the residuals. Finally, we utilize the ephemerality masks to finetune the 3D Gaussians.

and vehicles, to maintain robustness against pixels that represent transient entities. This necessitates addressing a robust optimization problem, where the outliers are transient objects—those that are either in motion or capable of moving—while the inliers are backgrounds.

## 4.2 Overall Architecture

Figure 2 shows our overall workflow. Given RGB images $\mathbf{I}$ collected across multiple traversals, we first leverage the classic Structure from Motion (SfM) [2] to jointly reconstruct sparse points for the initialization of Gaussians and obtain the camera poses $\boldsymbol{\xi} = \{\boldsymbol{\xi}_t \mid t = 1, \ldots, T\}$. We then utilize the differential rendering pipeline of 3DGS to learn the positions, rotations, scales, opacities, colors, and semantic features of the 3D environmental Gaussians $\mathbf{G}$, supervised by ground truth RGB $\mathbf{I}$ and self-supervised feature maps [4] denoted by $\mathbf{F} = \{\mathbf{F}_t \in \mathbb{R}^{w \times h \times d} \mid t = 1, \ldots, T\}$. Then we exploit the feature residual maps to extract ephemeral object masks denoted by $\mathbf{M} = \{\mathbf{M}_t \in \mathbb{R}^{w \times h} \mid t = 1, \ldots, T\}$. Finally, we finetune 3D Gaussians $\mathbf{G}$ through robust optimization by leveraging the ephemerality masks. In summary, 3DGM includes the three stages denoted by Initialization, EmerSeg, and EnvGS, as shown in Appendix A.1. We detail each stage from Sec. 4.3 to 4.5.

## 4.3 Initialization**: Structure from Motion**

The SfM pipeline frequently faces challenges in single-traversal scenarios, largely due to the limited scene coverage achieved with RGB observations collected along a narrow and long camera trajectory. Conversely, RGB images from multiple traversals offer a broader array of viewpoints, significantly improving the triangulation and bundle adjustment processes. Additionally, this approach can leverage the 2D consensus of hand-crafted features in the correspondence search, improving robustness against transient objects, which manifest as dissensus pixels across traversals. Moreover, our empirical experiments underscore the importance of the number of traversals for smooth initialization. A reduction in traversals can lead to a lack of sufficient image data, thereby failing the SfM initialization.

## 4.4 EmerSeg**: Emerged Ephemerality Segmentation by Feature Residuals Mining**

**Feature distillation** We utilize robust feature representations to enhance consensus verification, as the feature space exhibits better robustness against lighting variations and embodies semantic meanings, facilitating the decomposition of the transient objects by removing groups of semantically dissensus pixels. We minimize the following RGB and feature rendering loss:

$$\mathcal{L} = \sum_t (\mathcal{L}_{rgb}(\mathbf{I}_t(\boldsymbol{\xi}_t; \mathbf{G}), \mathbf{I}_t) + \mathcal{L}_{feat}(\mathbf{F}_t(\boldsymbol{\xi}_t; \mathbf{G}), \mathbf{F}_t)) \tag{2}$$

where $\mathbf{I}_t(\boldsymbol{\xi}_t; \mathbf{G}) \in \mathbb{R}^{w \times h \times 3}$ and $\mathbf{F}_t(\boldsymbol{\xi}_t; \mathbf{G}) \in \mathbb{R}^{w \times h \times d}$ are the rendered RGB image and feature map given pose $\boldsymbol{\xi}_t \in \mathfrak{se}(3)$ and Gaussians $\mathbf{G}$. $\mathbf{I}_t$ and $\mathbf{F}_t$ are the corresponding ground truth RGB and feature map. $\mathcal{L}_{rgb}$ and $\mathcal{L}_{feat}$ are loss functions for RGB images and semantic features. *As inlier pixels substantially outweigh outlier pixels, the model is primarily steered by gradients from*

*consensus inlier pixels towards learning permanent features. As a result, pixels manifesting high loss in feature space are very likely to be outliers.*

**Feature residuals mining**   We derive transient object masks by leveraging the spatial information in the feature residual maps, as shown in the right column of Fig. XVI~XXI. After training, we normalize the feature residuals and suppress pixels with residual values below a predefined threshold. Contours are then extracted from the normalized residual maps using spatial gradient information [68]. We refine these contours by applying spatial priors to eliminate those that are too small or located in the sky. Finally, we merge nearby contours and extract a convex hull for each merged contour. Ultimately, ephemerality masks $\mathbf{M}$ are produced from simple postprocessing of feature residuals without additional training. More details are shown in Appendix A.2.

### 4.5   `EnvGS`: Environmental Gaussian Splatting via Robust Optimization

After obtaining ephemerality masks $\mathbf{M}$, we focus on minimizing the following robust loss function (taking L1 loss as an example):

$$\mathcal{L} = \sum_t \mathcal{L}_{rgb}(\mathbf{M}_t \odot \mathbf{I}_t(\boldsymbol{\xi}_t; \mathbf{G}), \mathbf{M}_t \odot \mathbf{I}_t) \tag{3}$$

where $\mathbf{M}_t$ is an ephemerality mask for the $t$-th image to downgrade the influence of outlier pixels. Optionally, we employ a depth smoothness loss and sky masks to further improve the geometry reconstruction, as illustrated in Appendix A.3.

### 4.6   Comparison to Arts

The pioneering work addressing similar problems is NeRF-W [69], which learns volumetric representations from unconstrained photo collections. It employs uncertainty estimation to mask transient objects situated in image areas of high uncertainty. The following research efforts propose to learn a transient mask, aiming to eliminate occluders [70, 71]. Another related work is RobustNeRF [72] which models distractors in training data as outliers of an optimization problem and proposes a form of robust estimation for NeRF training.

We have three main differences from prior works.

- **Target problem**   We formulate robotic multitraverse RGB mapping as a robust representation learning problem, unlike previous works that focus on object-centric neural rendering of outdoor landmarks or multiple objects in indoor scenarios.
- **Scene decomposition**   Our method enables a clearer decomposition of foreground and background, producing both 2D segmentation and 3D environmental Gaussians without any supervision. This represents a significant improvement over previous methods, which produce only blurry results in outdoor scenarios.
- **Technical novelty**   We use Gaussian Splatting instead of the conventional NeRF approach. Our robust feature distillation and feature residuals mining fully exploit the spatial information of the rendering loss map, resulting in much better ephemerality segmentation.

## 5   Experiments

**Dataset**   Most NeRF benchmarks [38, 44, 46] for driving focus on a single-traversal video of the Waymo [73] or nuScenes [74]. To address the gap, we introduce the first *unsupervised **Map**ping and segmentation via multitra**verse*** (**Mapverse**) benchmark, which comprises **Mapverse-Ithaca365** (see Appendix B.1) and **Mapverse-nuPlan** (see Appendix B.2) derived from the Ithaca365 [5] and nuPlan [6] datasets, respectively. **Mapverse** features 40 locations, each with 10~16 traversals, yielding a total of 467 videos and 35,304 images. Due to space constraints, we present results for **Mapverse-Ithaca365** (20 locations, 200 videos, 20,000 images) in the main text, with additional results in **Mapverse-nuPlan** provided in Appendix F~H. Sample data are visualized in Figs. I~IV.

**Task and implementation**   We benchmark three tasks: *(1) unsupervised 2D ephemerality segmentation*, *(2) 3D reconstruction*, and *(3) neural rendering* in multitraversal driving. Our benchmark can

Table 1: **Mean IoU of unsupervised vs. five supervised methods in Mapverse-Ithaca365.** * indicates the model without training on our dataset.

| Sup. \ Unsup. | EmerSeg (Ours) | STEGO* [82] | STEGO [82] | CAUSE [83] |
|---|---|---|---|---|
| PSPNet [75] | **42.22** | 20.55 | 22.22 | 19.12 |
| SegViT [76] | **46.16** | 21.18 | 23.57 | 19.64 |
| InternImage [79] | **46.34** | 21.29 | 23.68 | 19.93 |
| Mask2Former [77] | **42.28** | 20.83 | 23.03 | 20.88 |
| SegFormer [78] | **45.14** | 21.31 | 23.78 | 20.63 |

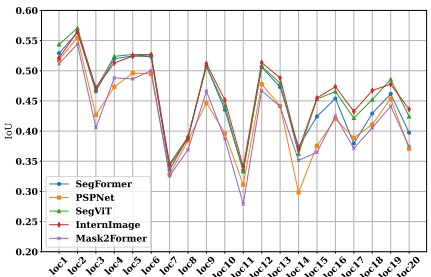

Figure 3: **IoU at 20 locations in Ithaca, NY.**

inspire wide applications in unsupervised perception, autolabeling, camera-only 3D reconstruction and neural simulation in self-driving and robotics. For efficiency, we compress feature dimensions from 768 to 64 using PCA. Our model uses KL divergence for feature alignment and L1 loss for RGB reconstruction. All experiments are conducted on a single NVIDIA RTX 3090 GPU.

## 5.1 Unsupervised 2D Ephemeral Object Segmentation

**Task setup**     Our `EmerSeg` can segment ephemeral traffic participants in a multitraverse image collection, *without any supervision*. This will help identify moving objects like vehicles and pedestrians, as well as static objects with the potential for movement, such as parked cars or traffic cones. We use a *training-as-optimization* pipeline and adopt the *Intersection over Union (IoU) metric* for evaluation. Regarding comparison methods, we employ several state-of-the-art *semantic segmentation* models trained with human annotations to create pseudo ground-truth masks for transient objects (pedestrians, vehicles, bicyclists, and motorcyclists). We also compare `EmerSeg` with *unsupervised segmentation* methods. We report the main comparison results in Sec. 5.1.1 and ablation studies in Sec. 5.1.2.

### 5.1.1 Quantitative and Qualitative Evaluations

**Comparison against supervised methods**     We compare our method with state-of-the-art (SOTA) semantic segmentation methods: PSPNet [75], SegViT [76], Mask2Former [77], SegFormer [78], and InternImage [79]. *Note that these methods require dense pixel-level annotations to learn semantics.* We directly use these models trained on either ADE20K [80] or Cityscapes [81] to produce masks on **Mapverse-Ithaca365**. The overall IoU scores of `EmerSeg` average around 0.45 compared to the five supervised models; see Tab. 1. IoU scores across 20 locations are detailed in Fig. 3, with seven locations surpassing 50% IoU, and the highest score reaching 56% compared to SegFormer. These results highlight the promising potential of our unsupervised segmentation paradigm.

**Comparison against unsupervised methods**     We compare `EmerSeg` with two SOTA unsupervised segmentation methods, *i.e.*, STEGO [82] and CAUSE [83]. We train both methods on our dataset using their unsupervised objectives. *Note that these unsupervised baseline methods cannot grasp the semantics or the concept of ephemerality and can only perform clustering within a single image.* Following prior work, we use a Hungarian matching algorithm to align the unlabeled clusters with pseudo ground-truth masks for evaluation. As shown in Tab. 1, `EmerSeg` significantly outperforms STEGO and CAUSE, with a 21.36-point (89.8%) IoU improvement over STEGO using SegFormer masks. More importantly, `EmerSeg` can understand ephemerality, a capability lacking in prior works.

**Qualitative comparison**     `EmerSeg` performs well in various lighting and weather conditions, effectively segmenting cars, buses, and pedestrians; see Fig. 4. However, it struggles with small or distant objects due to low feature map resolution. *We empirically find that small objects have minimal impact on neural rendering as they occupy few pixels.* Additional qualitative results are in Fig. V, with visualizations of baseline methods in Fig. VI. Detailed limitations are discussed in Appendix I.

**Computation time**     Figure VII illustrates the convergence of our segmentation method, showing a rapid increase in IoU during the initial iterations, which stabilizes around iteration 4,000. Notably, a resolution of 110×180 requires only 2,000 iterations to achieve an IoU score exceeding 40%, taking ∼8 minutes on a single NVIDIA RTX 3090 GPU for 1,000 images from 10 traversals of a location.

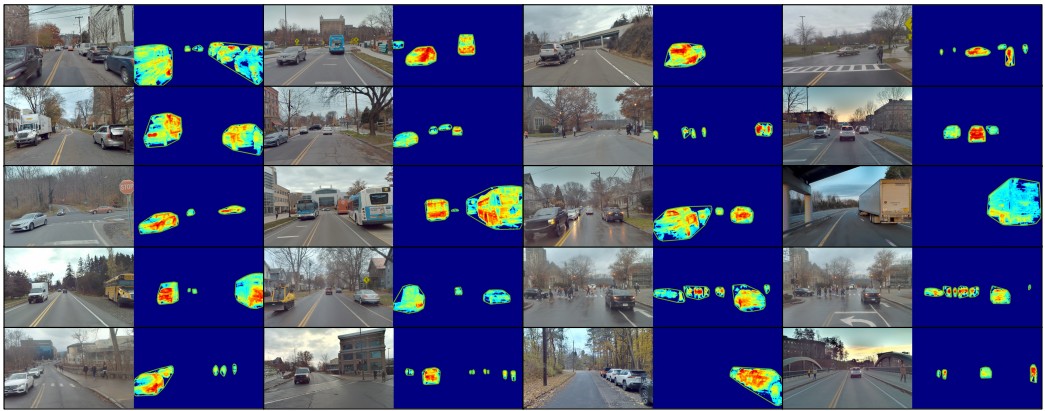

Figure 4: **Qualitative evaluations of** `EmerSeg` **in Mapverse-Ithaca365.**

Table 2: **Ablation Study Results of** `EmerSeg` **in Mapverse-Ithaca365.**

| Number of Traversals | | Feature Dimension | | | Feature Resolution | | | Feature Backbone | |
|---|---|---|---|---|---|---|---|---|---|
| # | IoU (%) | Dim. | Runtime | IoU (%) | Res. | Size (MB) | IoU (%) | Backbone | IoU (%) |
| 1 | 15.15 | 4 | 00:13:50 | 9.45 | 25×40 | 0.3 | 28.61 | DINO [54] | 16.51 |
| 2 | 42.31 | 8 | 00:16:50 | 10.91 | 50×80 | 1.0 | 35.91 | Denoised DINO [3] | 14.95 |
| 3 | 46.62 | 16 | 00:18:37 | 26.32 | 70×110 | 1.9 | 40.09 | DINOv2 [4] | 35.14 |
| 5 | 53.68 | 32 | 00:24:48 | 37.51 | 110×180 | 5.0 | **44.13** | Denoised DINOv2 [3] | **44.13** |
| 9 | 54.50 | 64 | 00:40:25 | **44.13** | 140×210 | 7.4 | 42.48 | DINOv2-reg [67] | 23.51 |
| 10 | **56.01** | 128 | 01:13:53 | 42.55 | 160×260 | 10.5 | 41.19 | Denoised DINOv2-reg [3] | 36.30 |

### 5.1.2 Ablation Studies

**Segmentation performance benefits from more traversals** We evaluate 2D segmentation on 100 images from a single traversal, using inputs from varying numbers of traversals; see Tab. 2. Starting at 15.15% with one traversal, the IoU jumps to 42.31% with two, and continues to rise: 53.16% at 8 and 56.01% at 10 traversals. This shows a clear trend of improving IoU with more traversals, with significant gains between 1 and 2. Detailed visualizations are in Fig. VIII.

**Effective segmentation requires 32 feature dimensions** We use PCA to compress the dimensions of DINOv2 features to save computation and storage. Our tests on segmentation performance at various dimensions revealed that 32 is an approximate threshold; IoU scores decrease significantly to around 10%-25% when the number of dimensions falls below 32, as shown in Tab. 2. Qualitative comparisons of different feature dimensions are demonstrated in Fig. IX.

**A resolution of 70×110 can achieve an IoU >40%** Table 2 shows IoU at various feature resolutions and sizes. IoU improves significantly as resolution increases from 25×40 (28.61%, 0.3 MB) to 110×180 (44.13%, 5.0 MB). However, higher resolutions like 140×210 and 160×260 result in slightly lower IoU scores of 42.48% and 41.19%, despite larger sizes. This indicates an optimal resolution at 110×180, balancing accuracy and efficiency. Visualizations at different resolutions are in Fig. X.

**Vision foundation model matters in unsupervised segmentation** We use robust features from self-supervised vision foundation models like DINO [54], DINOv2 [4], and DINOv2 with registers [67]. Additionally, we employ DVT [3] to reduce grid-like artifacts in ViT feature maps. As shown in Tab. 2, Denoised DINOv2 outperforms other models, highlighting the importance of robust, discriminative features for identifying transient clusters. Detailed visualizations are in Fig. XI.

## 5.2 3D Environment Reconstruction

**Task setup** Our `EnvGS` can extract 3D points from Gaussian Splatting, enabling the reconstruction of 3D environments from camera-only input while effectively ignoring transient objects across repeated traversals. We utilize a *training-as-optimization* pipeline and employ the *Chamfer Distance (CD) metric* for quantitative evaluation. For our comparison baseline, we use the state-of-the-art

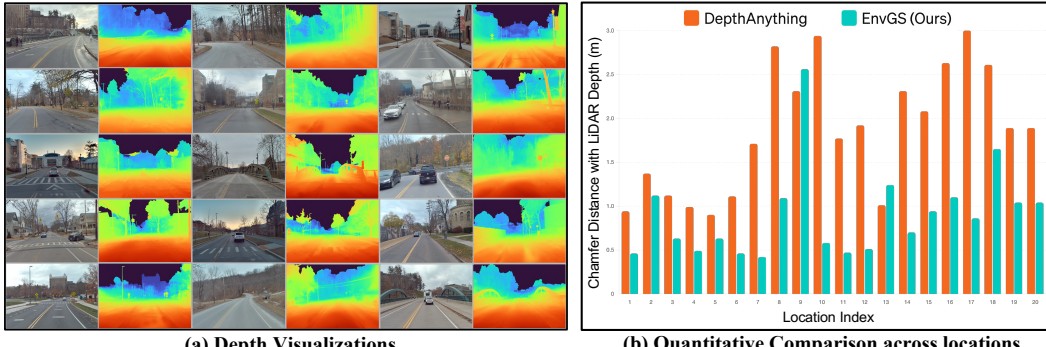

| (a) Depth Visualizations | (b) Quantitative Comparison across locations |

Figure 5: **Qualitative and quantitative evaluation of 3D geometry in Mapverse-Ithaca365.**

DepthAnything [84] model, which is trained with a combination of LiDAR ground truth (GT) depth data and unlabeled image data. This approach ensures that DepthAnything leverages diverse data sources to achieve satisfactory performance in zero-shot depth estimation.

**Quantitative results**    Figure 5 demonstrates the large reduction in Chamfer Distance (CD) achieved by `EnvGS` across nearly all locations. Our method achieves an average CD of approximately 0.9 meters, showcasing its precision in 3D reconstruction. Notably, there are five locations where the CD is even lower than 0.5 meters, highlighting the good accuracy of our approach in these areas. In contrast, DepthAnything has an average CD of around 1.9 meters, indicating a notable performance gap between the two methods. More importantly, our method avoids the need for costly LiDAR sensors during training, making it a cost-effective autonomous mapping solution for self-driving and robotics. Leveraging techniques such as mesh reconstruction [30] and 2D Gaussian Splatting [85] could further enhance the geometric reconstruction capabilities of our method.

**Qualitative results**    Figure 5 showcases depth visualizations of `EnvGS` across various driving scenarios. The depth maps generated by `EnvGS` exhibit superior accuracy, with smooth transitions from near to far objects and well-defined edges of scene structures. Additionally, `EnvGS` effectively removes transient objects without human supervision. Visualizations in 3D are shown in Fig. XII.

### 5.3  Neural Environment Rendering

**Task setup**    Our `EnvGS` can also achieve novel view synthesis through splatting-based rasterization. The challenge lies in ensuring the environment rendering automatically bypasses the non-environment pixels, *i.e.*, transient objects. We evaluate the quality of rendered images using three metrics: Learned Perceptual Image Patch Similarity (LPIPS), Structural Similarity Index (SSIM), and Peak Signal-to-Noise Ratio (PSNR). Given the absence of ground truth RGB images for clean backgrounds, we utilize the pretrained SegFormer [78] model to isolate foreground regions, allowing us to focus our evaluation exclusively on the quality of the background rendering.

**Baseline methods**    Our baseline methods include two NeRF-based methods, leveraging the implementation framework of iNGP [24]. The first, VanillaNeRF, constructs the scene within a single, static hash table and directly learns grid features from multitraverse images. In contrast, RobustNeRF [72] introduces an adaptive weighting mechanism to filter out outliers. In addition to the original 3DGS framework, we introduce two 3DGS-based baseline methods. 3DGS+RobustNeRF integrates the loss function from RobustNeRF, and 3DGS+SegFormer utilizes masks generated by a supervised segmentation model. For a fair comparison, all methods exclusively rely on camera images as input.

**Results and discussions**    Table 3 presents a quantitative comparison of various methods, showing that 3DGS-based approaches outperform NeRF-based methods. Adding the RobustNeRF loss function does not improve rendering quality in driving scenes. However, incorporating SegFormer or EmerSeg masks achieves the best LPIPS and SSIM. This is notable within a purely self-supervised framework, showcasing the potential of our self-supervised paradigm in pushing the boundaries of neural mapping. We present qualitative examples in Fig. 6, where it is evident that the original

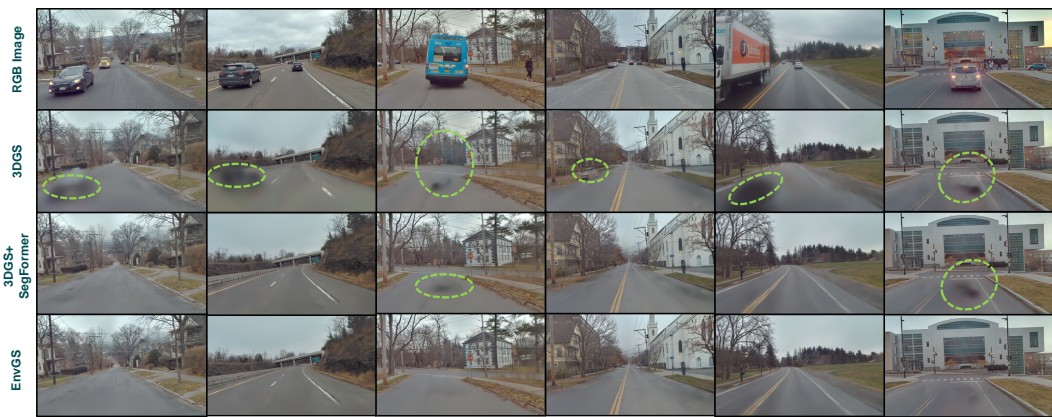

Figure 6: **Qualitative evaluations of the environment rendering.** Our method demonstrates robust performance against transient objects, and can even outperform the method equipped with a pretrained model in some cases. Notably, this includes the effective removal of object shadows.

Table 3: **Quantitative evaluation of novel view synthesis.** We set test/training views as 1/8. Pixels corresponding to transient objects are removed in the evaluations since we do not have ground truth background pixels in these regions occluded by transient objects.

| Metrics \ Methods | VanillaNeRF [24] | RobustNeRF [72] | 3DGS+RobustNeRF | 3DGS [1] | 3DGS+SegFormer | EnvGS (Ours) |
|---|---|---|---|---|---|---|
| LPIPS ($\downarrow$) | 0.423 | 0.443 | 0.416 | 0.227 | 0.212 | 0.213 |
| SSIM ($\uparrow$) | 0.603 | 0.609 | 0.654 | 0.798 | 0.806 | 0.806 |
| PSNR ($\uparrow$) | 19.18 | 19.22 | 19.97 | 22.92 | 22.81 | 22.78 |

3DGS model struggles with accurately reconstructing background regions affected by transient objects. More interestingly, our method can identify and mask out not only the objects themselves but also their associated non-environmental elements, such as shadows, as shown in the third and sixth columns of Fig. 6. More qualitative examples can be found in Fig. XIV.

## 6   Conclusion

**Broader impacts**   The concept of vision-only neural representation learning through repeated traversals extends beyond object segmentation and environment mapping, benefiting the vision and robotics communities. With a neural map prior, our approach becomes a powerful self-supervised framework for change detection and object discovery. This capability to render and analyze multitraverse environments over time is crucial for identifying environmental changes, aiding in early intervention for deforestation, urban expansion, or post-disaster assessments. Additionally, our method can serve as a baseline for autolabeling 2D masks and has potential for 3D autolabeling with LiDAR integration.

**Limitations**   Our method faces limitations in modeling large environmental variations, including nighttime conditions, major seasonal shifts, and adversarial weathers. We also note the presence of noise in the segmentation outputs caused by motion blur or appearance shifts. Leveraging temporal information or more powerful vision foundation models could help address this issue. More discussions can be found in Appendix I.

**Summary**   We introduce `3D Gaussian Mapping (3DGM)`, a novel self-supervised, camera-only framework that utilizes repeated traversals for simultaneous 3D environment mapping (`EnvGS`) and 2D unsupervised object segmentation (`EmerSeg`). Additionally, we develop the **Mapverse** benchmark, comprising nearly 500 driving video clips from the Ithaca365 and nuPlan datasets. Our method's effectiveness in unsupervised 2D segmentation, 3D reconstruction, and neural rendering is validated through both qualitative and quantitative assessments in repeated driving scenarios. Furthermore, `3DGM` opens new research opportunities, such as online unsupervised object discovery and offline autolabeling. We believe our work will advance vision-centric and learning-based self-driving and robotics, setting new standards in multitraverse setups and self-supervised scene understanding.

## Acknowledgement

We express our deep gratitude to Jiawei Yang and Sanja Fidler for their valuable feedback throughout the project. We also thank Yurong You and Carlos A. Diaz-Ruiz for their support with the Ithaca365 dataset, and Shijie Zhou for his help with high-dimensional feature rendering in 3DGS. Yiming Li gratefully acknowledges support from the NVIDIA Graduate Fellowship Program.

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

# Contents (Appendix)

# Appendix

## A  `3DGM`: Additional Details

### A.1  Workflow of `3DGM`

- **Stage-1:** `Initialization` with COLMAP.
    - Input: RGB images $\mathbf{I}$
    - Output: A sparse set of 3D points and camera poses $\boldsymbol{\xi}$.
- **Stage-2:** `EmerSeg`: Ephemerality Segmentation via Feature Residuals Mining.
    - Input: RGB images $\mathbf{I}$, semantic feature maps $\mathbf{F}$, camera poses $\boldsymbol{\xi}$.
    - Output: 2D ephemerality masks $\mathbf{M}$.
- **Stage-3:** `EnvGS`: Environmental Gaussian Splatting via Robust Optimization.
    - Input: RGB images $\mathbf{I}$, ephemerality masks $\mathbf{M}$, camera poses $\boldsymbol{\xi}$.
    - Output: 3D environmental Gaussians $\mathbf{G}$.

### A.2  Workflow of Feature Residuals Mining

**Algorithm 1** Workflow of Feature Residuals Mining

---

**Input:** Feature residuals $\{\mathcal{L}_{feat}(\mathbf{F}_t(\boldsymbol{\xi}_t; \mathbf{G}), \mathbf{F}_t)\}_{t=1,2,...,T}$, activation threshold $\delta_1 = 0.3$, size threshold $\delta_2 = 100$, skyline threshold $\delta_3 = 0.7$, merging threshold $\delta_4 = 10$, and default parameters for contour detection.

**Output:** Ephemeral objects masks $\{\mathbf{M}_t\}_{t=1,2,...,T}$.

1: **for** each $t$ **do**
2:      Load feature residual map $\mathcal{L}_{feat}(\mathbf{F}_t(\boldsymbol{\xi}_t; \mathbf{G})$.
3:      Normalize the feature residual map over all pixels
4:      **Activation**. Set all pixels with values less than $\delta_1$ to zero.
5:      **Contour detection**. Use `cv.findContours()` function in OpenCV to retrieve contours from the activated feature residual map using the algorithm [68].
6:      **Small contours filtering**: Remove very small contours, which may result from noise features caused by motion blur or lighting changes, based on $\delta_2$.
7:      **Sky contours filtering**. Remove contours located in the sky based on $\delta_3$.
8:      **Contours merging**. Merge nearby contours according to the threshold $\delta_4$. Merging helps create a more coherent and accurate outline of objects, especially when they are segmented into multiple smaller contours due to noise or slight variations in pixel values.
9:      **Extract a convex hull for each merged contour**. A convex hull provides a simplified representation of the shape by enclosing all the points of the contour with the smallest convex polygon. This makes the shape easier to process and analyze. Meanwhile, convex hull extraction helps smooth out irregularities and minor indentations in the contour, leading to a more uniform and stable shape.
10:     **Mark pixels inside convex hulls as masked-out regions**.
11: **end for**

## A.3 Additional Loss Function

We offer an optional geometry-related loss function to enhance depth reconstruction when the focus is more on geometry than photometry.

**Inverse Depth Smoothness Loss**    This loss function [86] encourages the smoothness of the depth map in non-edge areas with the penalty on the disparity gradients $\nabla D_{i,j}$. Using image gradients $\nabla I_{i,j}$ as weights reduce the impact of the loss in regions where edges are present, maintaining depth discontinuities at edges. The loss is formulated as:

$$\mathcal{L}_{depth} = \frac{1}{N} \sum_{i,j} \left( |\nabla D_{i,j}^x| \exp\left(-\|\nabla I_{i,j}^x\|\right) + |\nabla D_{i,j}^y| \exp\left(-\|\nabla I_{i,j}^y\|\right) \right) \tag{4}$$

where D represents the inverse of the rendered depth map, and I is the ground truth image.

**Sky Loss**    We aim to manipulate the opacity of the sky to 0 and other areas in the image to 1.

$$\mathcal{L}_{sky} = \frac{1}{N} \sum_{i,j} \left( |\mathcal{M}_{sky} - (1 - \mathcal{O})| \right) \tag{5}$$

where $\mathcal{M}_{sky}$ is the sky mask, with values of 1 for sky pixels and 0 for others, and $\mathcal{O}$ denotes the rendered opacity ranging from 0 to 1.

# B The Mapverse Dataset

We curate the Mapverse dataset based on Ithaca365 [5] and nuPlan [6]. Ithaca365 emphasizes its multitraverse nature in the original paper, whereas nuPlan does not explicitly mention this feature. These two datasets capture diverse scenes to verify our method across various driving scenarios. Both datasets use the Creative Commons Attribution-NonCommercial-ShareAlike 4.0 International Public License ("CC BY-NC-SA 4.0"). The configuration of our Mapverse dataset is shown in Table I. Further details are discussed below.

## B.1 Mapverse-Ithaca365

The Ithaca365 dataset [5] collects 40 traversals along a 15 km route under diverse scenarios, spanning the period from August 2021 through March 2022. The main goal of Ithaca365 is to develop robust perceptual systems for various weather conditions, including snow and rain. We select a subset of 10 traversals with similar weather conditions (7 cloudy, 2 sunny, and 1 rainy day) for our purposes. The specific dates are 11-19-2021, 11-22-2021, 11-30-2021, 12-01-2021, 12-06-2021, 12-07-2021, 12-14-2021, 12-15-2021, 12-16-2021, and 01-16-2022, most of which lie within one month (from mid-November to mid-December), with only one collection in mid-January. Meanwhile, we segment each long video sequence into multiple 20-second clips, with each clip capturing a specific location. Ultimately, Mapverse-Ithaca365 features 20 locations, each associated with 10 traversals. Each traversal contains 100 images, yielding a total of 20,000 images (200 videos). Some example data from 20 locations are shown in Fig. I and Fig. II. Note that each row features a different location, while each column represents a different traversal.

## B.2 Mapverse-nuPlan

The nuPlan dataset [6] is a comprehensive dataset designed to advance research and development in autonomous vehicle planning. Developed by Motional, it is considered the world's first and largest benchmark for AV planning. The dataset includes approximately 1,500 hours of driving data collected from four cities: Boston, Pittsburgh, Las Vegas, and Singapore. These cities were chosen for their unique driving challenges, such as bustling casino pick-up and drop-off points in Las Vegas. The authors provide 10% of the raw sensor data (120 hours). We find that the nuPlan dataset collected in Las Vegas has a number of repeated traversals of the same location. Hence, we extract the multitraverse driving data (from mid-May to late July 2021) by querying the GPS coordinates and curate our Mapverse-nuPlan dataset with a total of 20 locations, 267 videos, and ~15,000 images. Some example data from 20 locations are shown in Fig. III and Fig. IV. Note that each row features a different location, while each column represents a different traversal.

Table I: **Details of the Mapverse Dataset.**

| Mapverse-Ithaca365 | | | | Mapverse-nuPlan | | | |
|---|---|---|---|---|---|---|---|
| Location Index | # of Traversals | # of Images | # of Cameras | Location Index | # of Traversals | # of Images | # of Cameras |
| 1 | 10 | 1000 | 1 | 1 | 13 | 818 | 1 |
| 2 | 10 | 1000 | 1 | 2 | 13 | 778 | 1 |
| 3 | 10 | 1000 | 1 | 3 | 14 | 793 | 1 |
| 4 | 10 | 1000 | 1 | 4 | 12 | 710 | 1 |
| 5 | 10 | 1000 | 1 | 5 | 13 | 767 | 1 |
| 6 | 10 | 1000 | 1 | 6 | 13 | 588 | 1 |
| 7 | 10 | 1000 | 1 | 7 | 14 | 760 | 1 |
| 8 | 10 | 1000 | 1 | 8 | 14 | 789 | 1 |
| 9 | 10 | 1000 | 1 | 9 | 13 | 759 | 1 |
| 10 | 10 | 1000 | 1 | 10 | 13 | 756 | 1 |
| 11 | 10 | 1000 | 1 | 11 | 13 | 777 | 1 |
| 12 | 10 | 1000 | 1 | 12 | 13 | 827 | 1 |
| 13 | 10 | 1000 | 1 | 13 | 14 | 733 | 1 |
| 14 | 10 | 1000 | 1 | 14 | 15 | 753 | 1 |
| 15 | 10 | 1000 | 1 | 15 | 14 | 788 | 1 |
| 16 | 10 | 1000 | 1 | 16 | 13 | 883 | 1 |
| 17 | 10 | 1000 | 1 | 17 | 13 | 756 | 1 |
| 18 | 10 | 1000 | 1 | 18 | 16 | 727 | 1 |
| 19 | 10 | 1000 | 1 | 19 | 13 | 740 | 1 |
| 20 | 10 | 1000 | 1 | 20 | 11 | 802 | 1 |
| **Total** | **200** | **20,000** | | | **267** | **15,304** | |

## B.3 Visualization of Sample Data

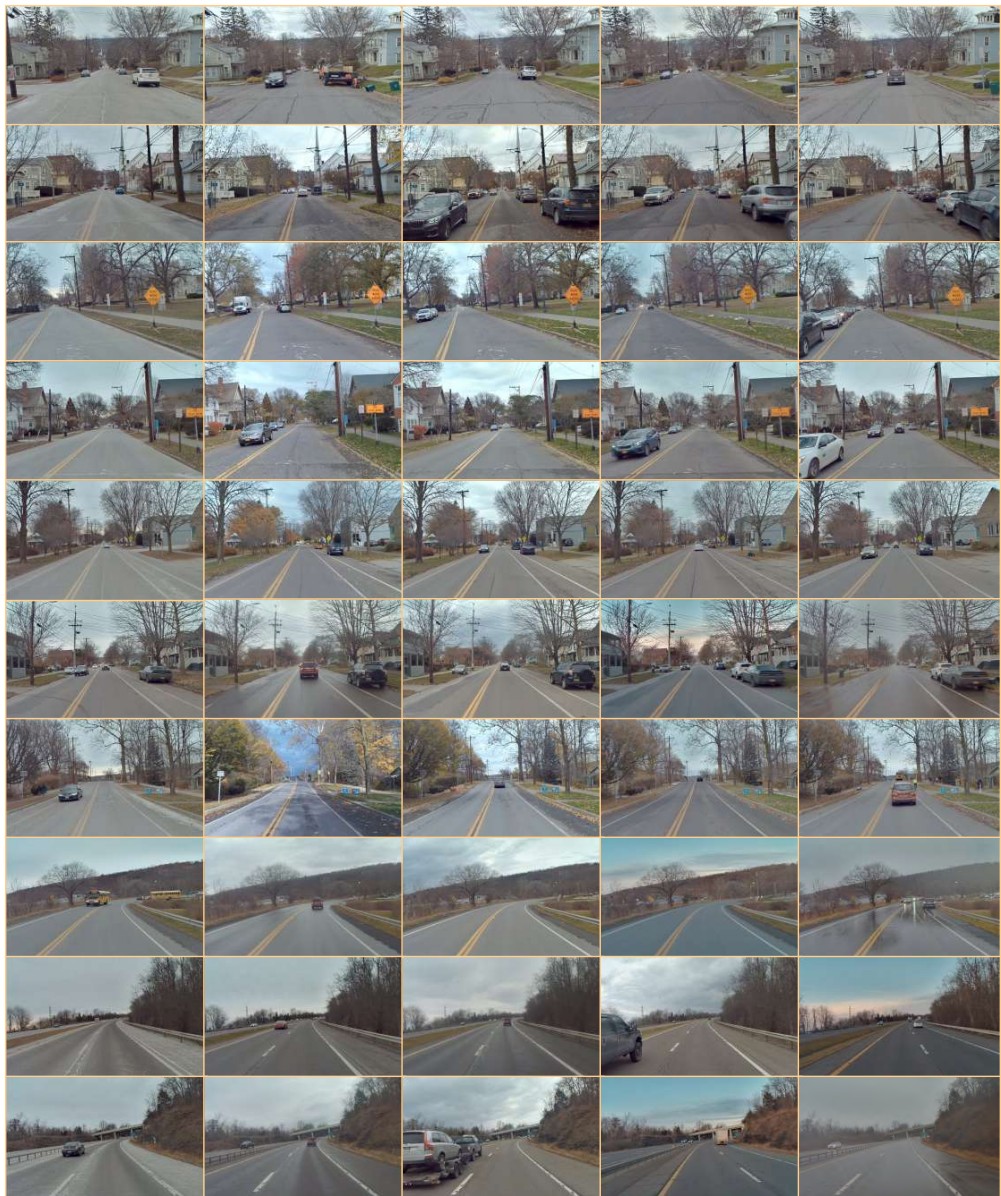

Figure I: **Visualizations of Mapverse-Ithaca365 dataset (locations 1-10).** Each row represents image observations of the same location captured during different traversals, with five traversals shown for brevity. The figure encompasses diverse environments in the Ithaca area, from residential neighborhoods with houses, trees, and varying traffic, to suburban streets with signage and seasonal foliage changes, and finally to rural roads and highways with expansive landscapes. The columns provide comparative views of these locations under different conditions, highlighting the dynamic nature of the Mapverse-Ithaca365 dataset.

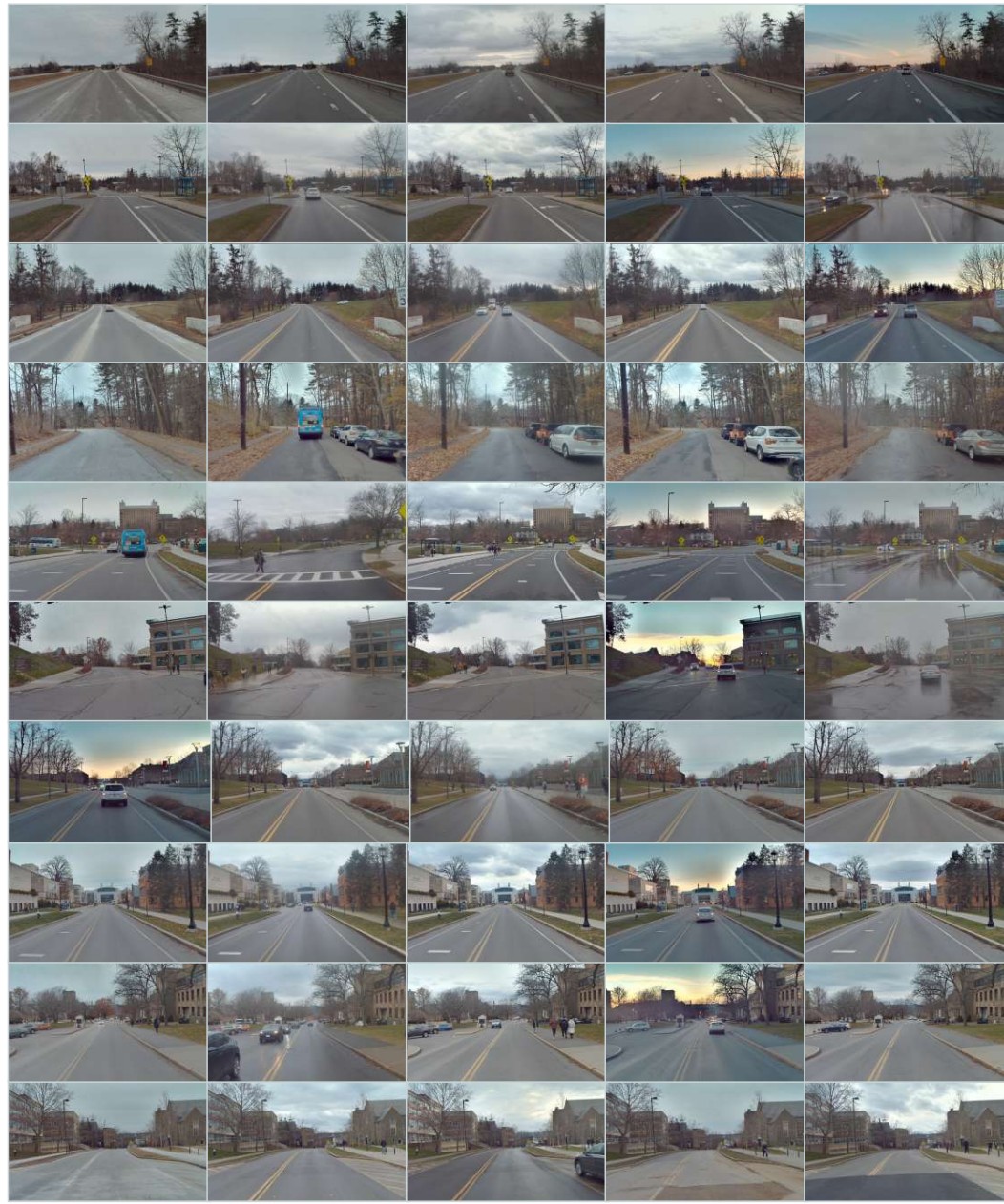

Figure II: **Visualizations of Mapverse-Ithaca365 dataset (locations 11-20).** Each row captures image observations of the same location from different traversals, showing five traversals for brevity. The figure spans various environments within Ithaca, from expansive rural highways transitioning to suburban roads with clear signage, to wooded areas with parked vehicles, and urban intersections with notable buildings. The images depict the progression from rural outskirts to more densely populated urban centers, reflecting changes in traffic, lighting, and seasonal foliage. Columns provide comparative views of these locations under different conditions, emphasizing the dynamic and diverse nature of the Mapverse-Ithaca365 dataset.

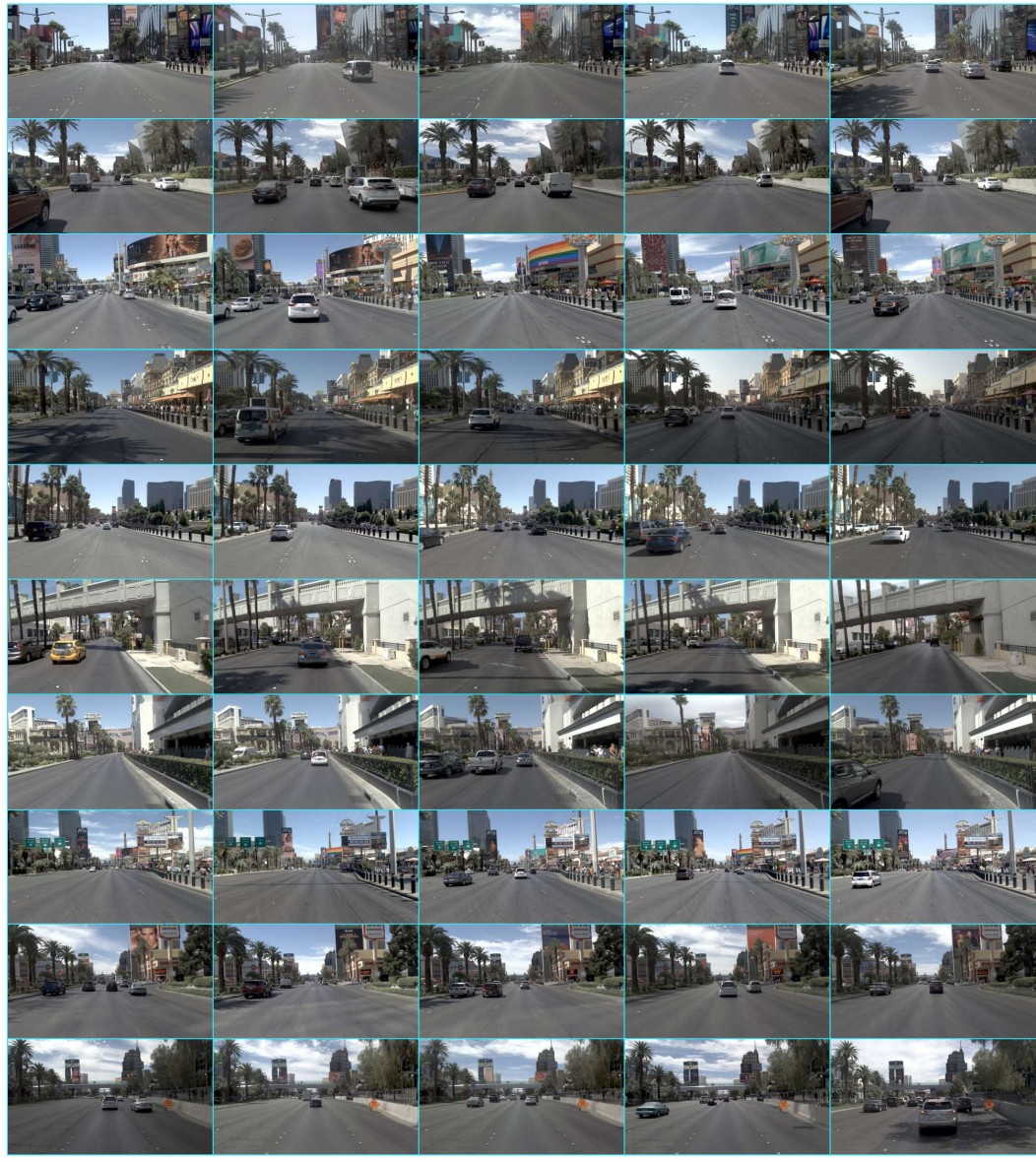

Figure III: **Visualizations of Mapverse-nuPlan dataset (locations 1-10).** Each row represents different image observations of the same location captured during multiple traversals, with five shown for brevity. The images encompass diverse environments in Las Vegas, including wide city streets with iconic buildings, billboards, palm trees, pedestrian bridges, and varying traffic conditions. Columns provide comparative views of the same locations under different conditions, illustrating the variability and complexity of the cityscape as captured in the Mapverse-nuPlan dataset.

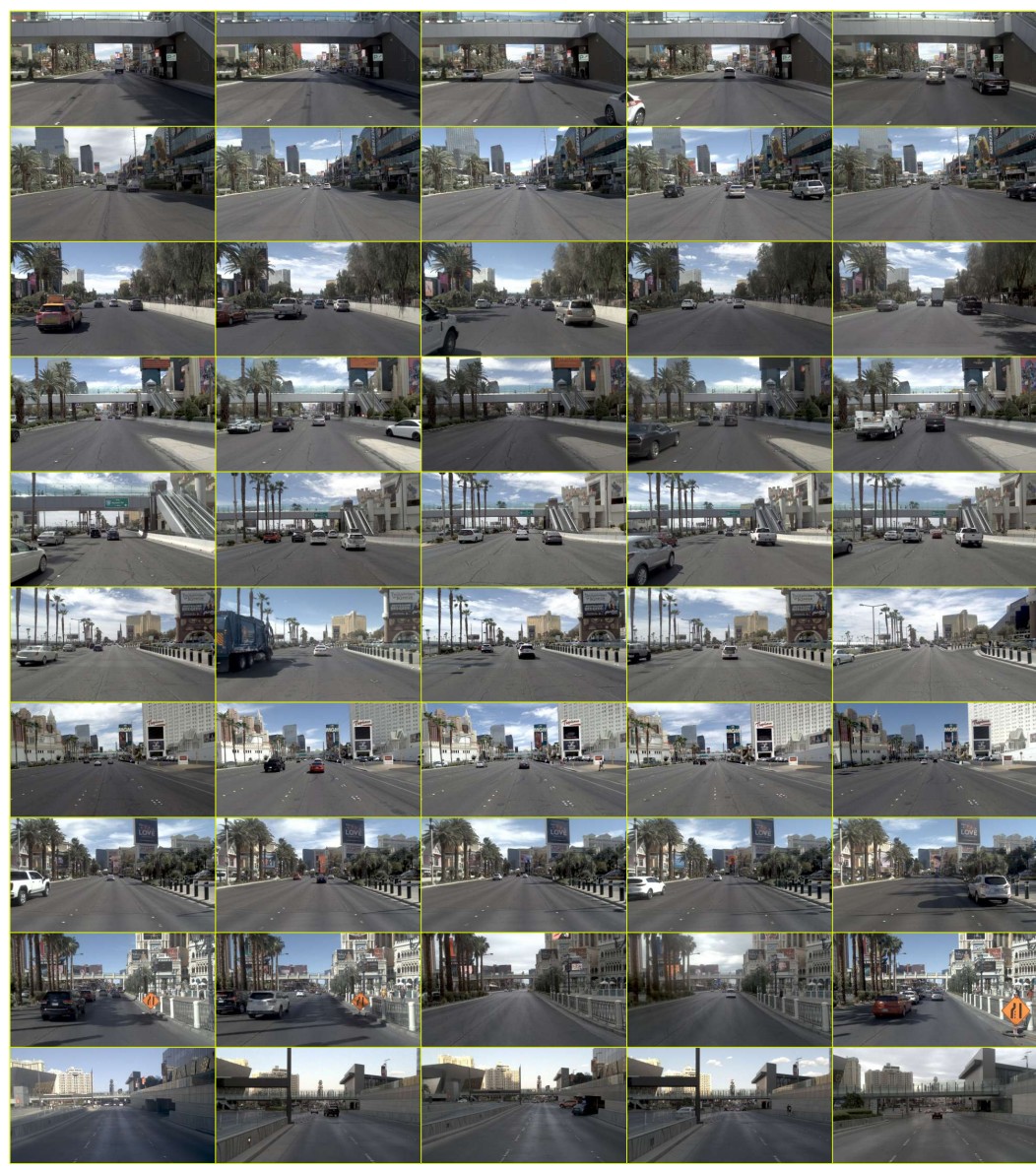

Figure IV: **Visualizations of Mapverse-nuPlan dataset (locations 11-20).** Each row represents different image observations of the same location captured during multiple traversals, with five shown for brevity. The images cover various environments in Las Vegas, including city streets with overpasses, iconic buildings, palm trees, billboards, and varied traffic conditions. The sequence progresses from urban settings with heavy infrastructure and prominent landmarks to broader streets and intersections, capturing different times of day and lighting conditions. Columns provide comparative views of the same locations under different circumstances, showcasing the dynamic and ever-changing urban landscape of Las Vegas as recorded in the Mapverse-nuPlan dataset.

# C Mapverse-Ithaca365: Additional Results of 2D Segmentation

## C.1 Additional Qualitative Results

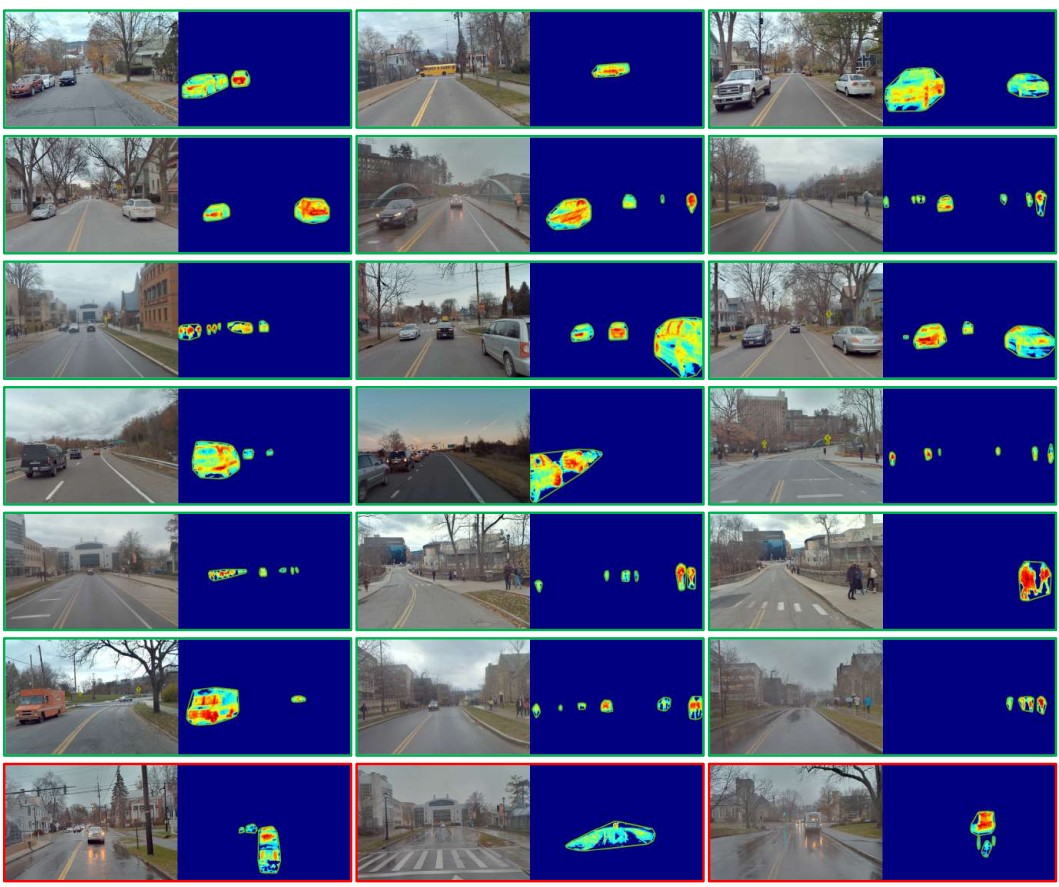

Figure V: **Qualitative evaluations of the emerged object masks**. Our method demonstrates robust performance across a range of lighting and weather conditions, effectively handling diverse categories including cars, buses, and pedestrians. Some failure cases are highlighted with red rectangles.

## C.2   Visualizations of Supervised and Unsupervised Segmentation

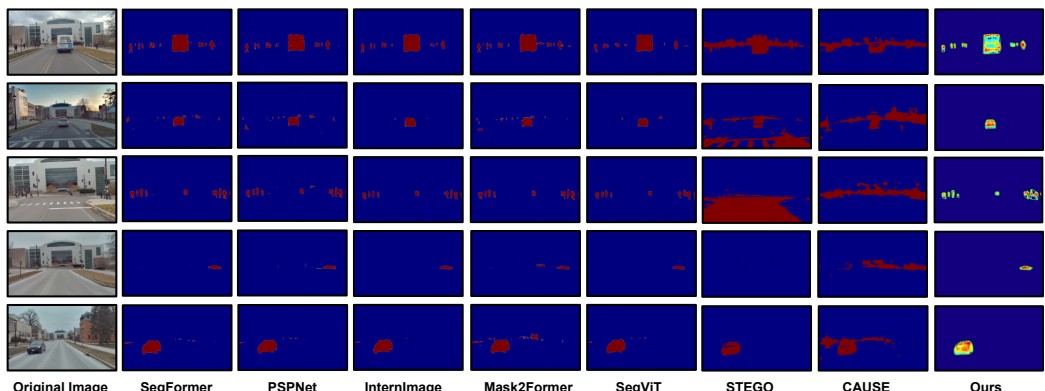

Figure VI: **Qualitative comparisons of our method and other supervised and unsupervised segmentation baselines.** This image demonstrates a comparison between our mask extraction and those derived from other semantic segmentation methods. The results indicate that our masks maintain superior integrity and detail in complex environments. Meanwhile, our method significantly outperforms unsupervised semantic segmentation models [82, 83] and is roughly equivalent to the masks generated by InternImage [79] and SegVit [76]. Although Mask2Former [77], PSPNet [87], and SegFormer [78] have advantages in recognizing people and other fine-grained objects, they can also lead to incorrect segmentation and noise in certain scenarios.

## C.3 Performance over Training Iterations

Figure VII presents the IoU performance across iterations for two different feature resolutions (110×180 and 140×210), alongside visualizations of ephemerality masks and feature residuals at various iterations. The IoU graph on the left shows that both resolutions exhibit rapid improvement in the initial iterations, with the 110×180 resolution consistently outperforming the 140×210 resolution. The 110×180 resolution reaches an IoU of approximately 0.44, while the 140×210 resolution plateaus around 0.41. This indicates that the lower resolution (110×180) is more efficient in capturing ephemeral objects. On the right, the visualizations of ephemerality masks and feature residuals at different iterations (500 to 10000) demonstrate that higher iterations result in more detailed and accurate segmentation. Early iterations (500 and 1000) show sparse and less accurate masks. The progression also highlights the fast convergence of our method for effective segmentation.

**Summary** In summary, the figure demonstrates that the 110×180 feature resolution is more effective and efficient for segmentation, achieving higher IoU scores compared to the 140×210 resolution. The IoU increases rapidly in the initial iterations and stabilizes around iteration 4000. These results emphasize the importance of selecting an appropriate feature resolution and ensuring sufficient iterations to achieve optimal segmentation performance.

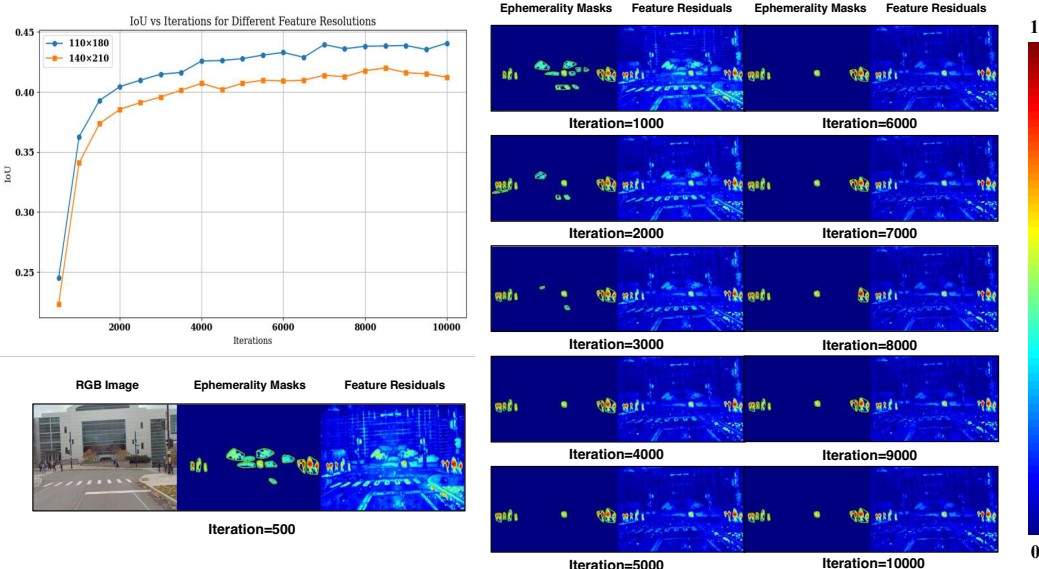

Figure VII: **IoU performance over iterations for different feature resolutions (110×180 and 140×210) and corresponding visualizations of ephemerality masks and feature residuals.** Visualizations at various iterations (500 to 10000) illustrate that higher iterations lead to more detailed and accurate segmentation. The results highlight the efficiency of the 110×180 resolution and the fast convergence of our method for effective segmentation.

## C.4 Ablation Study on Number of Traversal: Visualization and Discussion

Figure VIII showcases the segmentation performance of `EmerSeg` with images collected from varying numbers of traversals. Each row represents a different scene, while the columns illustrate the results from 1, 2, 3, 7, and 10 traversals, respectively. The segmentation map with a single traversal shows minimal detection of ephemeral objects, indicating limited information for effective segmentation. For example, the method fails to segment three parked buses in the first row due to a lack of diverse visual observations, which are crucial for our model to localize these transient yet static objects. With 2 traversals, there is a significant improvement in segmentation performance. The segmentation map reveals larger and more distinct objects, demonstrating the benefit of additional traversals. Segmentation performance with 10 traversals is similar to that with 7 traversals. Objects are detected reliably, but the improvement beyond 7 traversals is marginal, indicating diminishing returns.

**Summary** Figure VIII illustrates the clear trend of improving segmentation performance with an increasing number of traversals. The results indicate that while significant gains are achieved with additional traversals, the benefits plateau after a certain point. This analysis underscores the importance of multiple traversals for effective segmentation while suggesting an optimal balance between the number of traversals and segmentation accuracy.

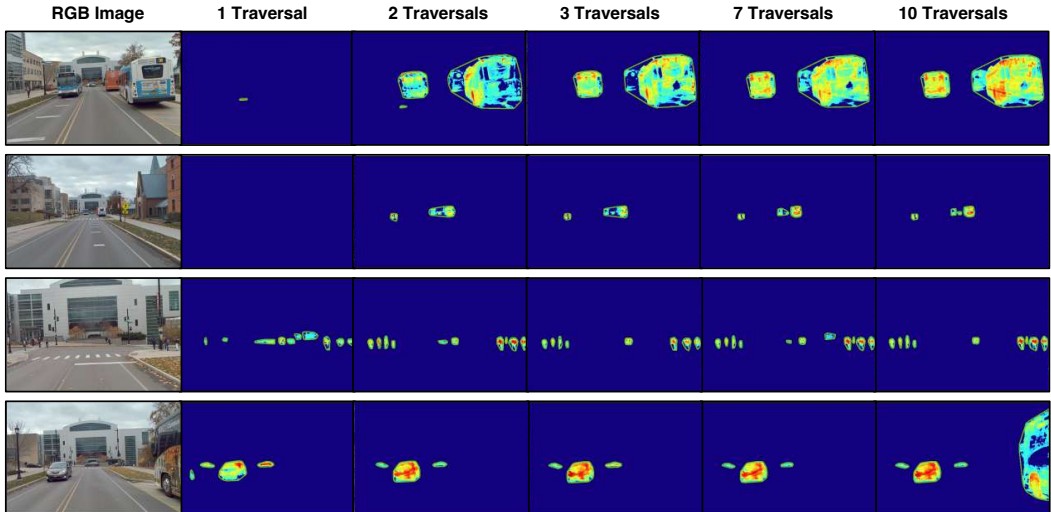

Figure VIII: **Visualizations of `EmerSeg` with inputs from different numbers of traversals.** Each row represents a different scene of a location. The first column shows the original RGB images. The subsequent columns show the segmentation outputs from `EmerSeg` with 1, 2, 3, 7, and 10 traversals.

## C.5 Ablation Study on Feature Dimension: Visualization and Discussion

Figure IX illustrates the impact of varying feature dimensions on the segmentation performance. At the lowest dimension (4), the ephemerality masks are sparse, capturing very few objects with minimal detail since the feature residuals are not discriminative, indicating insufficient feature representation. A substantial enhancement is observed at dimension 16, where the ephemerality masks become more detailed, capturing more objects with better clarity. At the highest dimension (64), the segmentation is highly detailed and accurate, with ephemerality masks capturing a wide range of objects and feature residuals being informative, suggesting a comprehensive feature representation.

**Summary** In summary, the segmentation performance improves significantly with increased feature dimensions. Low-dimensional features (4 and 8) fail to provide adequate information for accurate segmentation, resulting in sparse segmentation. A higher dimension (16 and 64) offers a substantial improvement, capturing most objects with clearer details. This demonstrates that higher-dimensional features are crucial for achieving accurate and comprehensive segmentation performance.

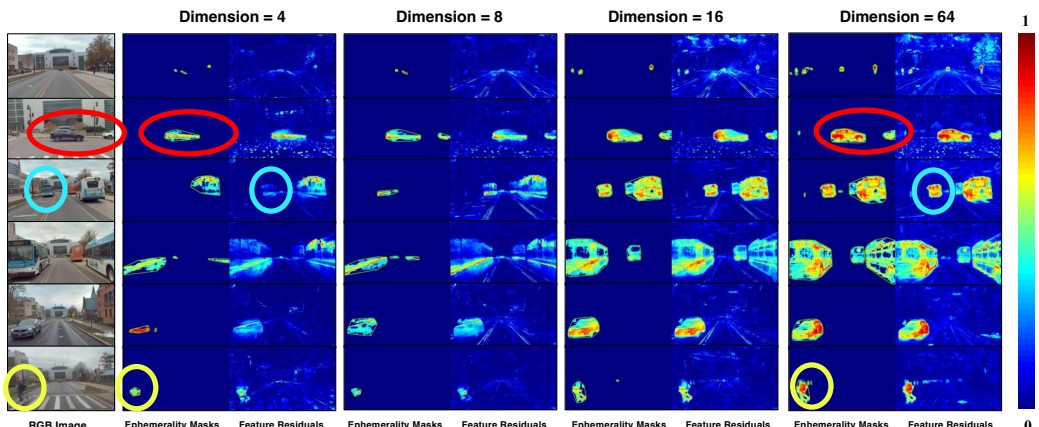

Figure IX: **Visualizations of ephemerality masks and feature residuals at different feature dimensions.** The RGB images (leftmost column) are processed to generate ephemerality masks and feature residuals. As the feature dimensions increase, the segmentation accuracy improves, with the highest dimension (64) capturing the most detailed and accurate object masks. The residuals are more discriminative with higher dimensions, indicating better feature representation. The colored circles highlight specific areas to illustrate differences in segmentation quality across dimensions.

## C.6 Ablation Study on Feature Resolution: Visualization and Discussion

Figure X presents the segmentation performance at different feature resolutions: 25×40, 50×80, 70×110, and 110×180. The RGB images on the left are segmented into ephemerality masks and feature residuals for each resolution. At the lowest resolution (25×40), the ephemerality masks capture very few objects with minimal detail. As the resolution increases, there is a noticeable improvement in object segmentation, with more objects being identified.

**Summary**  In a word, the segmentation performance improves significantly with increased feature resolution. Lower resolutions (25×40 and 50×80) result in sparse object segmentation, indicating inadequate feature representation. The highest resolution (110×180) delivers the best results, with detailed and precise segmentation.

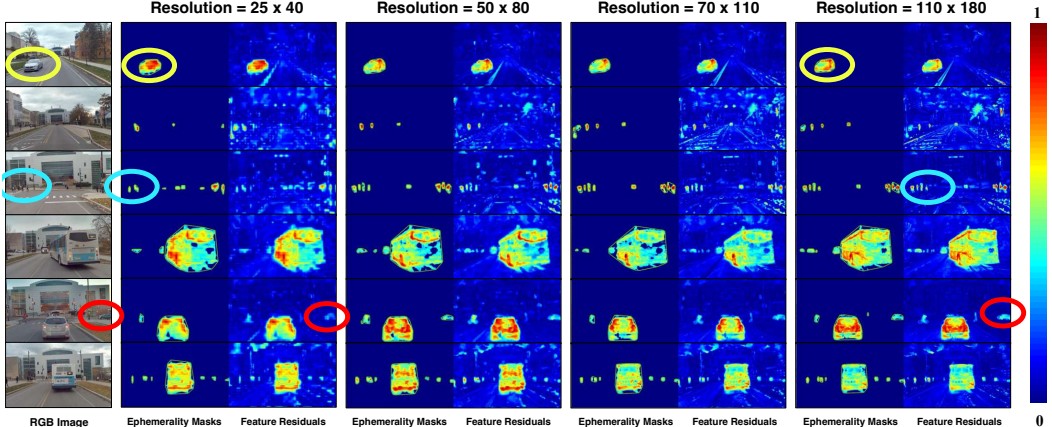

Figure X: **Visualizations of ephemerality masks and feature residuals at different feature (spatial) resolutions.** The RGB images (leftmost column) are processed to generate ephemerality masks and feature residuals. As the feature resolution increases, the segmentation accuracy improves, with the highest resolution (110×180) capturing the most detailed and accurate object masks. The residuals are more informative with higher resolutions, indicating better feature representation and reduced segmentation errors. The colored circles highlight specific areas to illustrate differences in segmentation quality across resolutions.

### C.7 Ablation Study on Vision Foundation Model: Visualization and Discussion

Figure XI compares the performance of different versions and configurations of the DINO model on ephemerality masks and feature residuals. For both DINOv1 and DINOv2, the raw versions show noisy feature residuals, indicating areas where the model fails to capture ephemeral objects accurately. The denoised versions show a notable improvement, with informative residuals and more accurate ephemerality masks. The DINOv2 models generally perform better than DINOv1, as evidenced by more precise object masks and more discriminative residuals. The inclusion of the register in DINOv2 does not introduce additional gains.

**Summary** The result highlights the progressive improvements in segmentation accuracy achieved through model evolution from DINOv1 to DINOv2, and the benefits of applying denoising techniques.

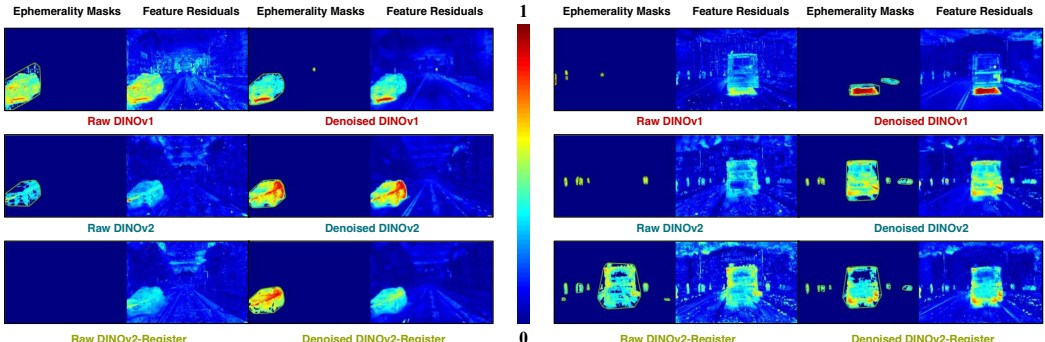

Figure XI: **Comparison of ephemerality masks and feature residuals using different versions of the DINO model.** The figure includes raw and denoised versions of DINOv1 and DINOv2, as well as raw and denoised versions of DINOv2 with a registration module (DINOv2-Register). Denoising enhances the quality of feature residuals, while registration does not yield notable gains.

# D    Mapverse-Ithaca365: Additional Visualizations of 3D Reconstruction

Figure XII compares Structure from Motion (SfM) initialized points with Gaussian points after optimization. On the left, the images display the raw data points obtained from the SfM process, which serves as an initial guess for the 3D structure of the environment. These points tend to be more scattered and less organized. On the right, the images show the Gaussian points after optimization, where the point cloud data has been refined through differentiable rendering. This refinement results in a more coherent and precise representation of the scene, as evidenced by the clearer and more defined structures in the point clouds. The comparison across various scenes highlights the effectiveness of the optimization process in enhancing the accuracy and clarity of the 3D reconstructed environment, crucial for applications in autonomous driving and robotics.

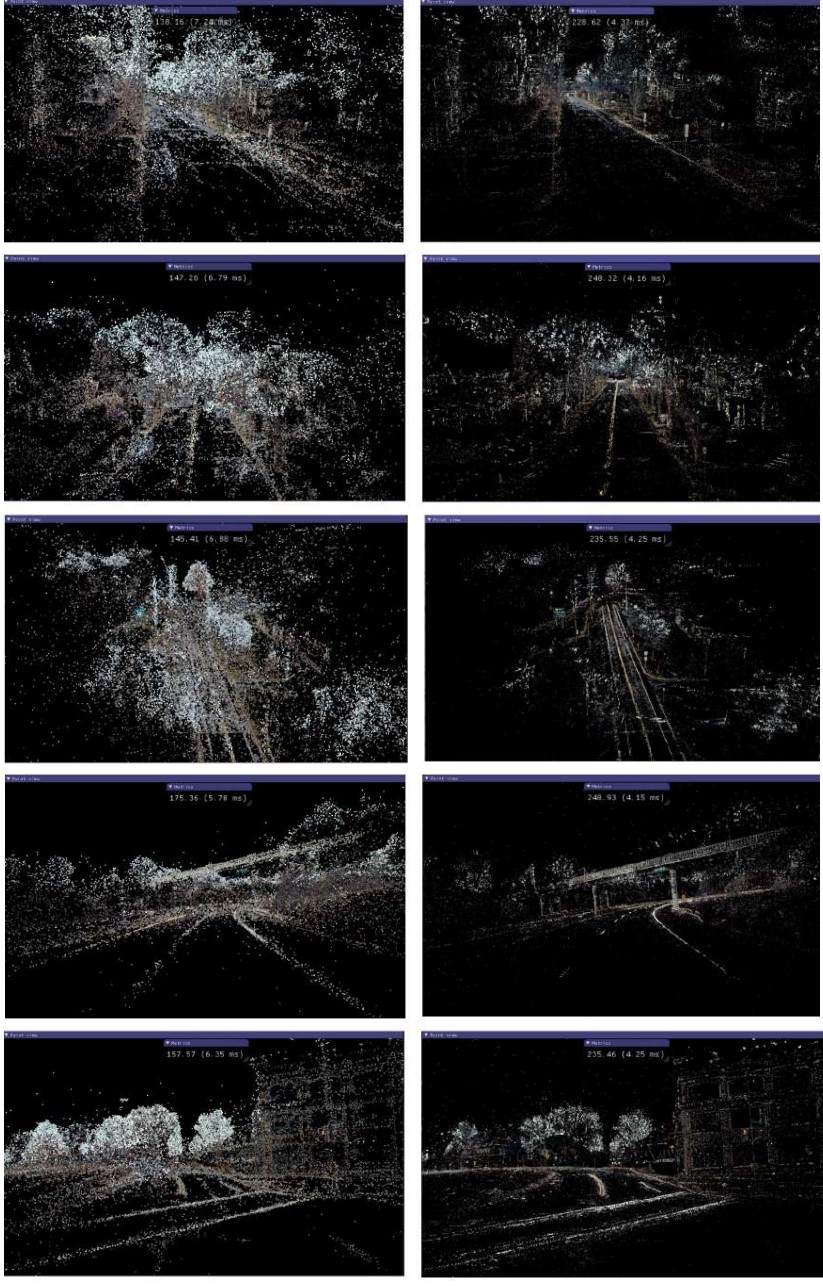

Figure XII: **Left: SfM Initialized Points. Right: Gaussian Points after Optimization.**

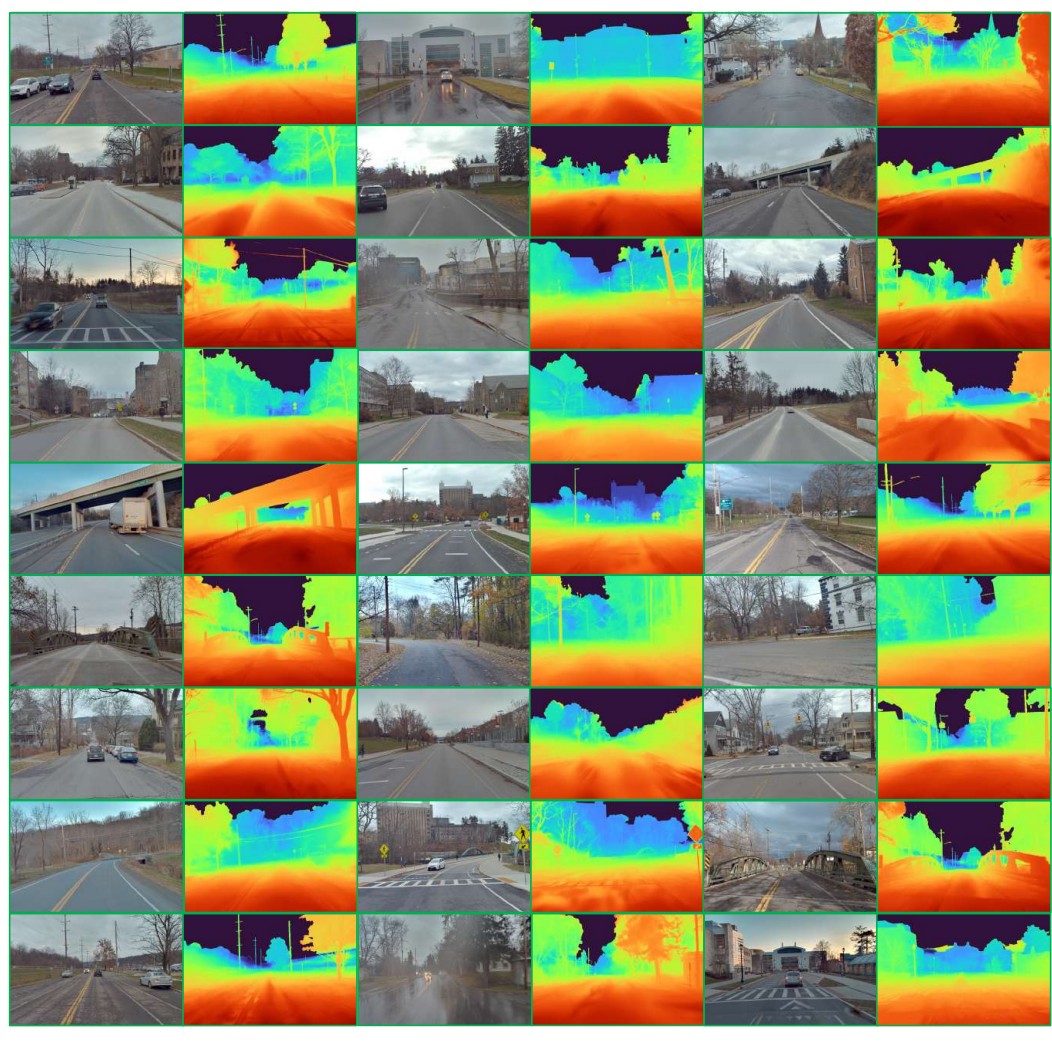

Figure XIII: **Visualizations of depth images in Mapverse-Ithaca365**

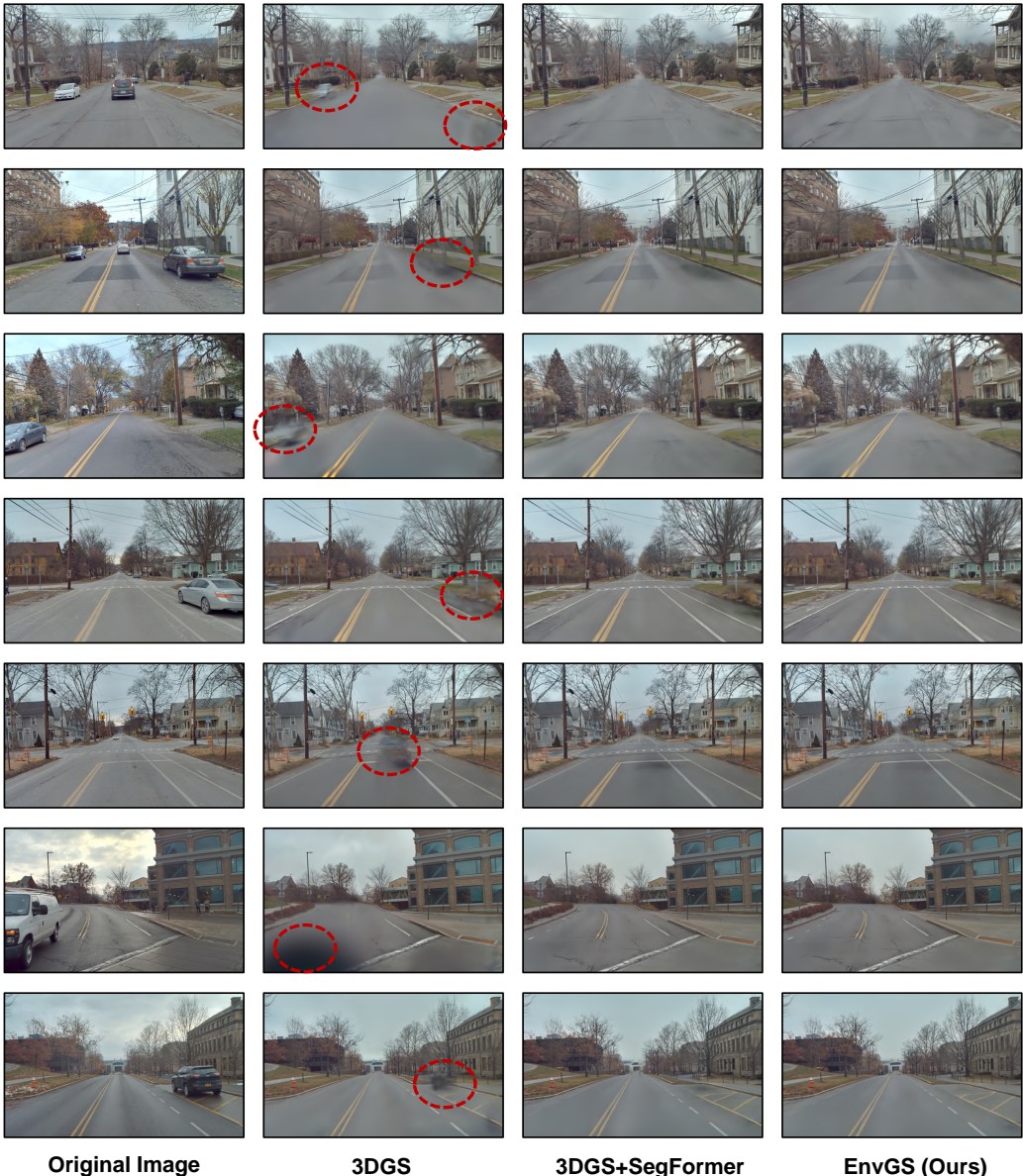

**Original Image**  **3DGS**  **3DGS+SegFormer**  **EnvGS (Ours)**

Figure XIV: **Qualitative evaluations of the environment rendering.** Our method demonstrates robust performance against transient objects.

# F  Mapverse-nuPlan: Unsupervised 2D Segmentation

## F.1  Quantitative Results

We employ SegFormer [78] to generate pseudo ground-truth masks and compare these with the output of `EmerSeg` using Intersection over Union (IoU) metrics in Mapverse-nuPlan collected in Las Vegas. Figure XV displays the IoU scores across different locations. The highest IoU score is at loc6 with 59.53%, indicating the best segmentation performance. In contrast, loc20 has the lowest score at 28.69%, indicating the poorest performance. **The average IoU score across all locations is approximately 46.51%**, which surpasses the IoU score of 45.14% on Mapverse-Ithaca365. Las Vegas is known for its dense traffic and complex urban environment compared to Ithaca, which presents a different set of challenges for segmentation models. The improved performance in the more demanding Las Vegas environment indicates that `EmerSeg` can *adapt to various traffic densities and urban complexities*, maintaining high accuracy without requiring extensive parameter tuning.

The variation in IoU scores across different locations can be attributed to several factors, including the complexity of the scene, lighting conditions, and the presence of occlusions. Locations with higher IoU scores, such as loc6, likely benefit from clearer images and less occlusion, allowing for more accurate segmentation. Conversely, locations like loc20, with lower IoU scores, may suffer from challenging conditions such as poor lighting and complex backgrounds.

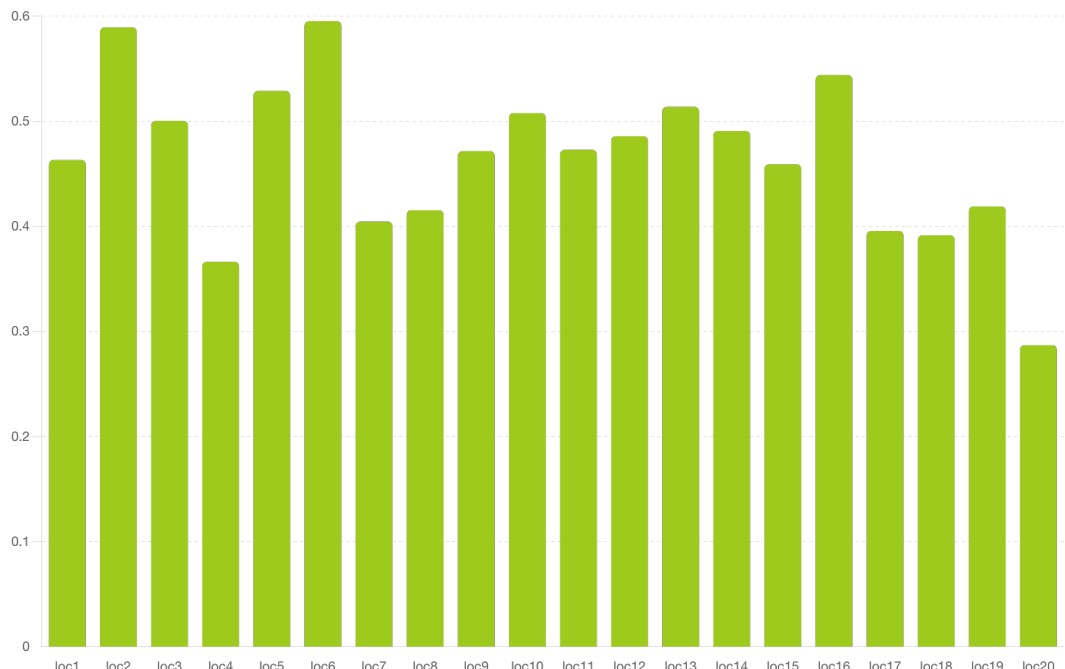

Figure XV: **IoU of** `EmerSeg` **compared to SegFormer across locations in Mapverse-nuPlan.**

## F.2  Qualitative Results

We visualize some segmentation results in Fig.XVI to Fig.XXI. The visualizations highlight `EmerSeg`'s ability to accurately segment objects in complex traffic situations with multiple dynamic elements. These results underscore the versatility and reliability of our approach, showcasing its potential for real-world applications in autonomous driving and other vision-based tasks.

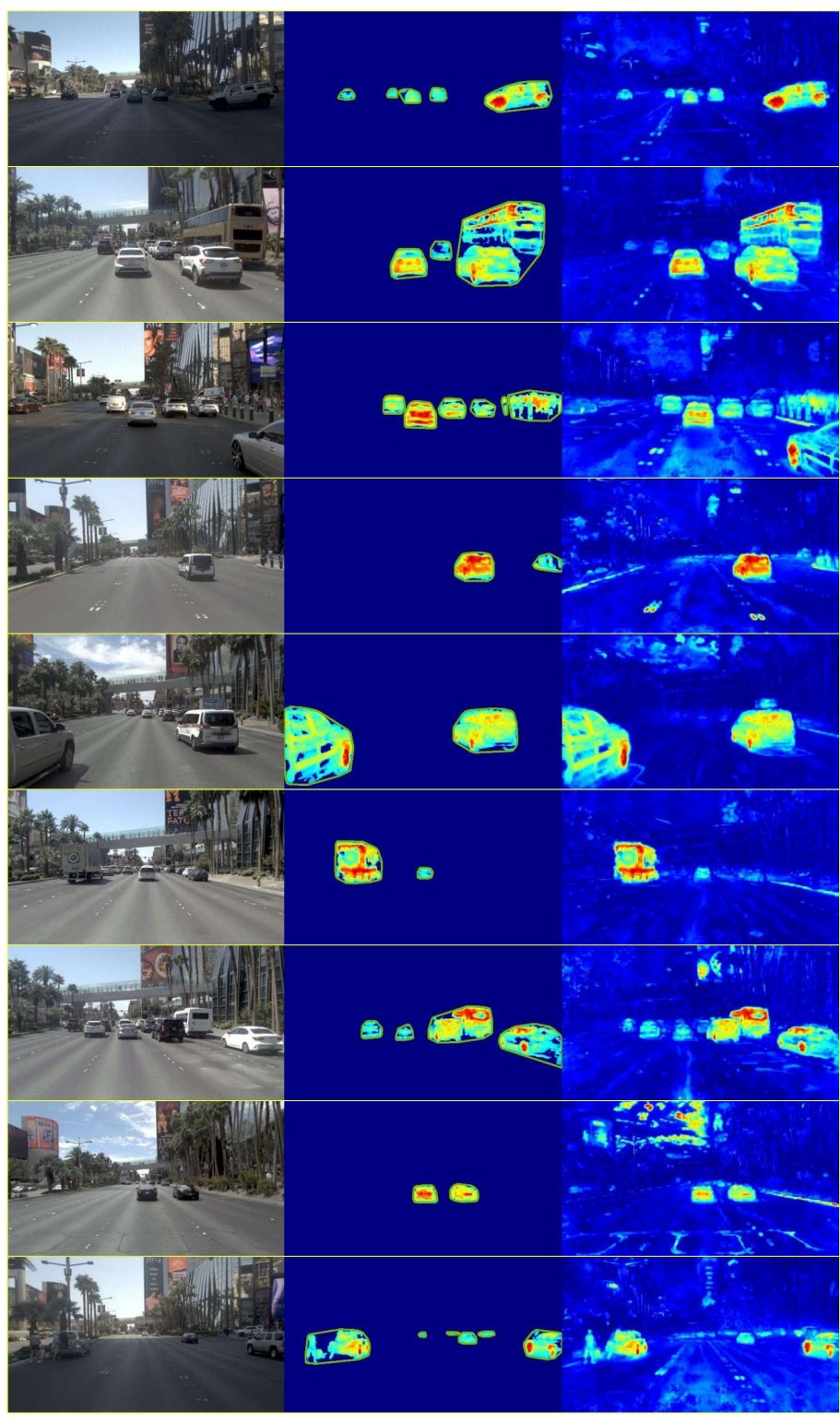

Figure XVI: **Qualitative results of** `EmerSeg` **for multiple traversals of location 1 of Mapverse-nuPlan.** From left to right: raw RGB image, extracted 2D ephemeral object masks, and normalized feature residuals visualized using a jet color map.

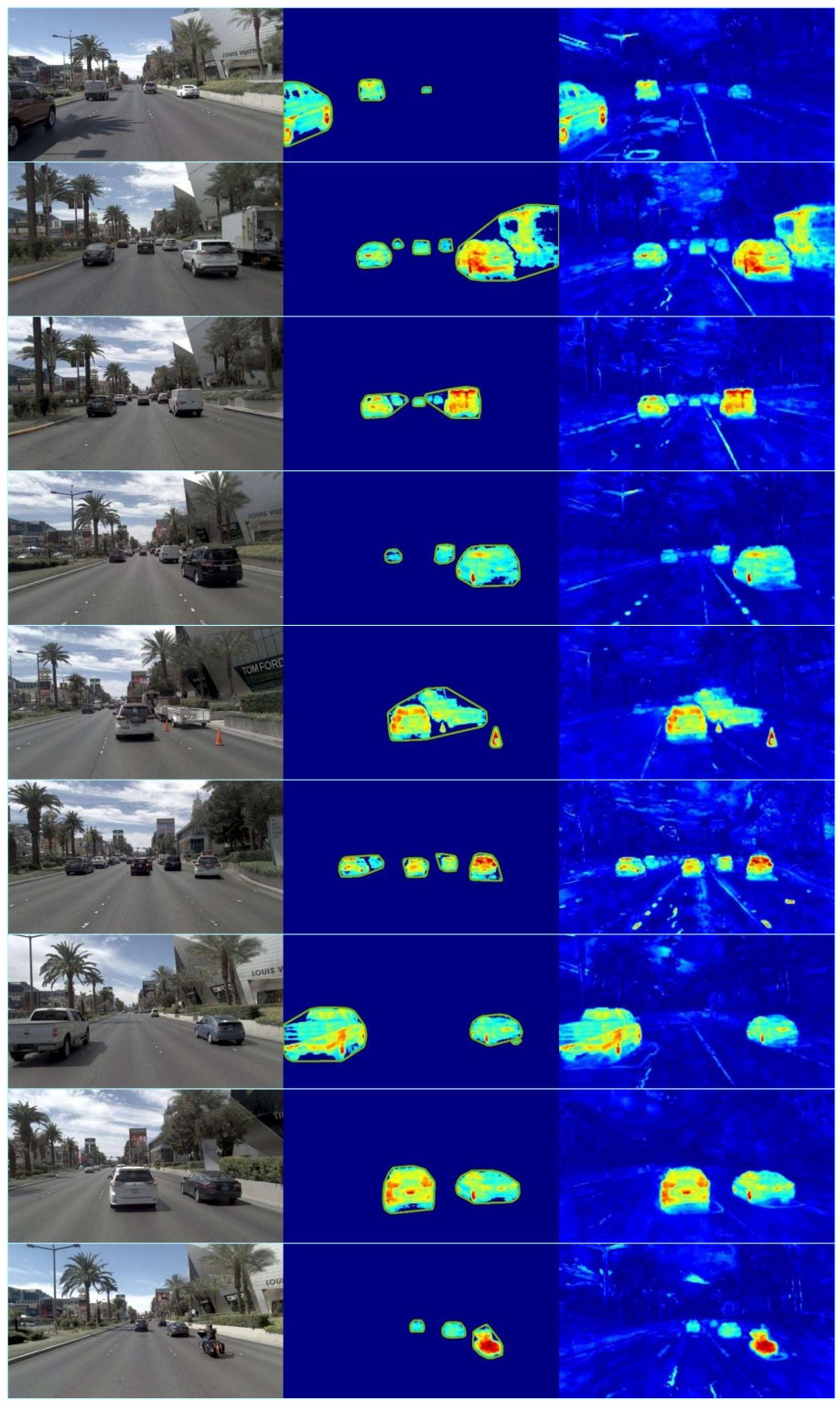

Figure XVII: **Qualitative results of** `EmerSeg` **for multiple traversals of location 2 of Mapverse-nuPlan.** From left to right: raw RGB image, extracted 2D ephemeral object masks, and normalized feature residuals visualized using a jet color map.

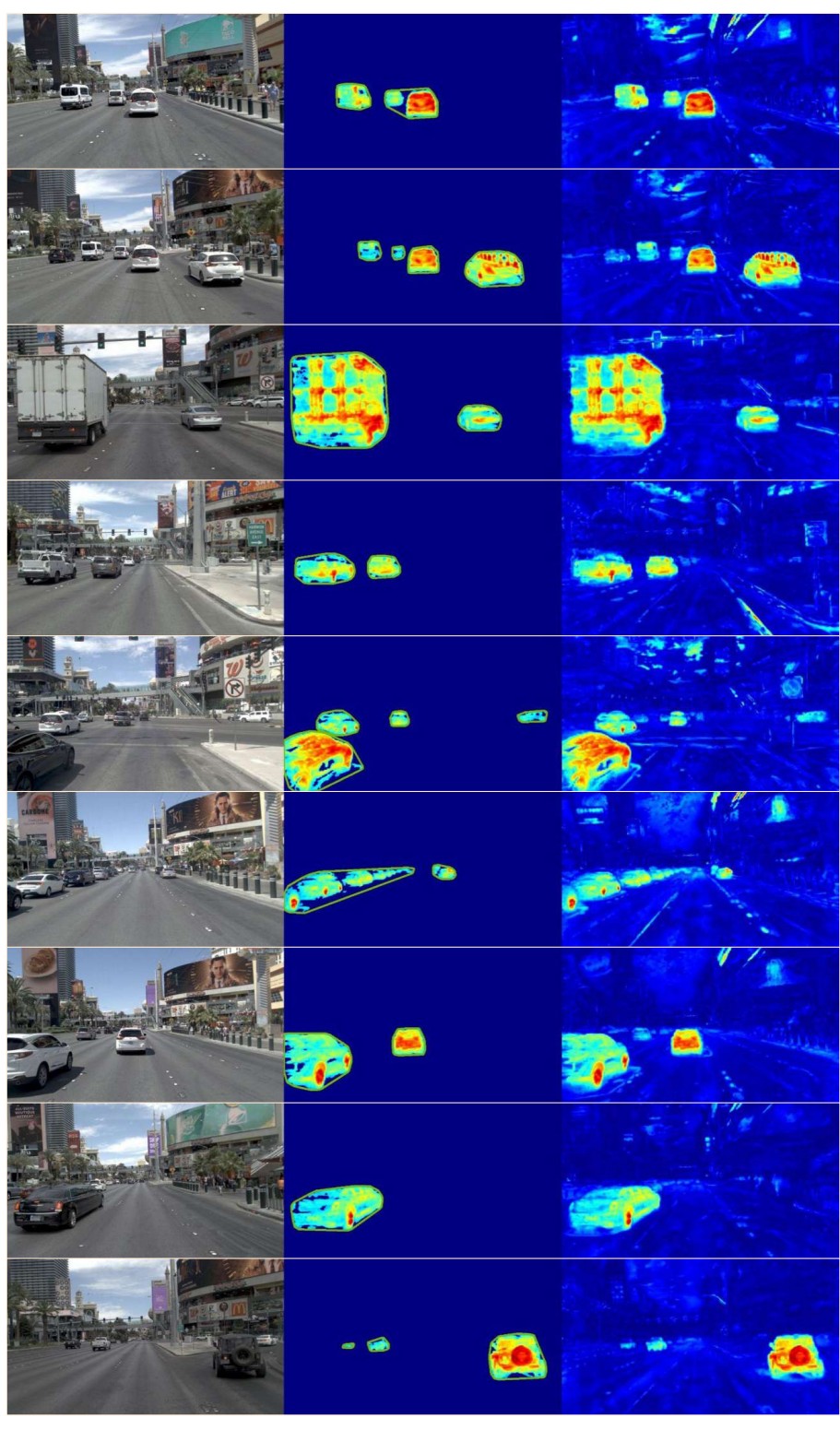

Figure XVIII: **Qualitative results of** `EmerSeg` **for multiple traversals of location 3 of Mapverse-nuPlan.** From left to right: raw RGB image, extracted 2D ephemeral object masks, and normalized feature residuals visualized using a jet color map.

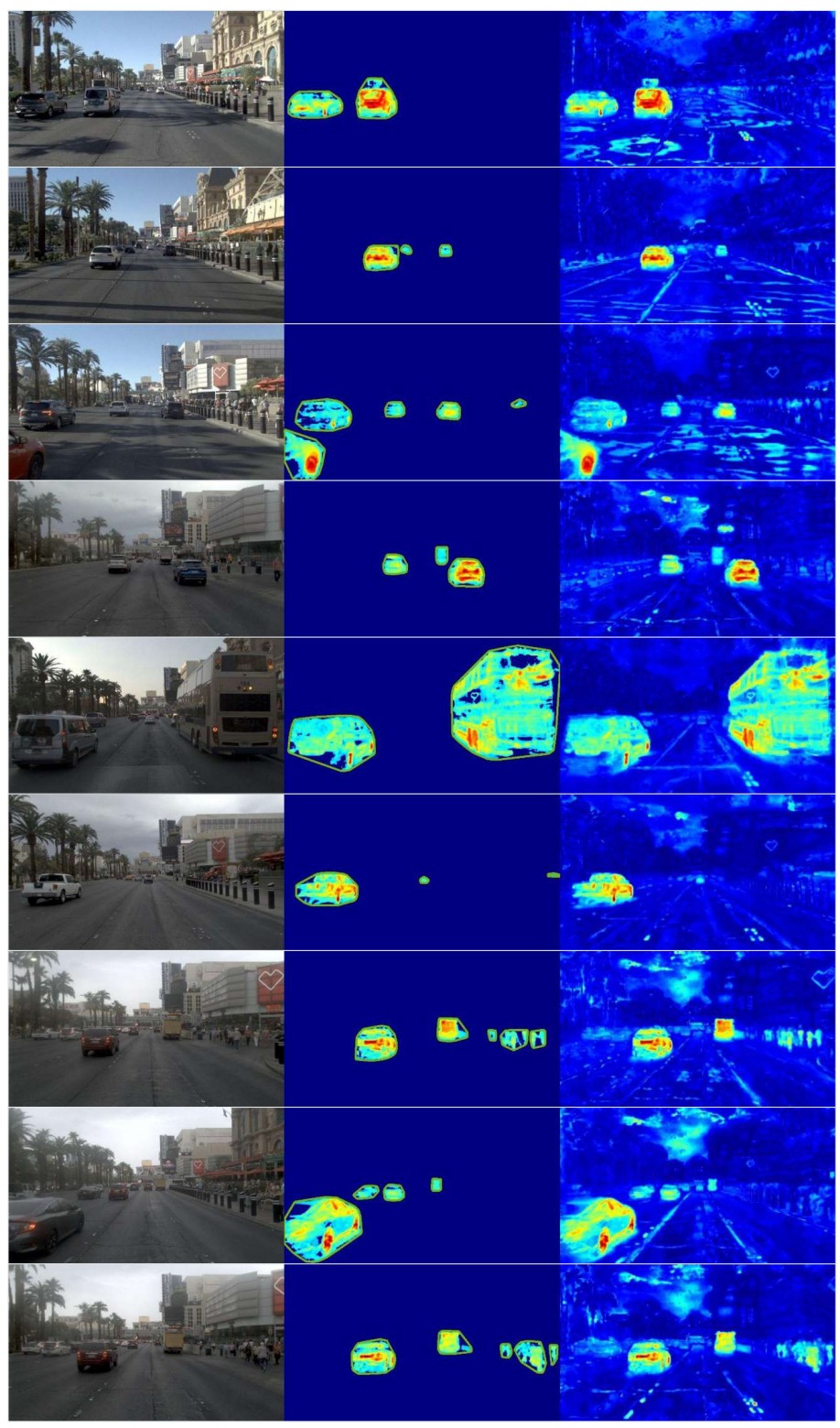

Figure XIX: **Qualitative results of** `EmerSeg` **for multiple traversals of location 4 of Mapverse-nuPlan.** From left to right: raw RGB image, extracted 2D ephemeral object masks, and normalized feature residuals visualized using a jet color map.

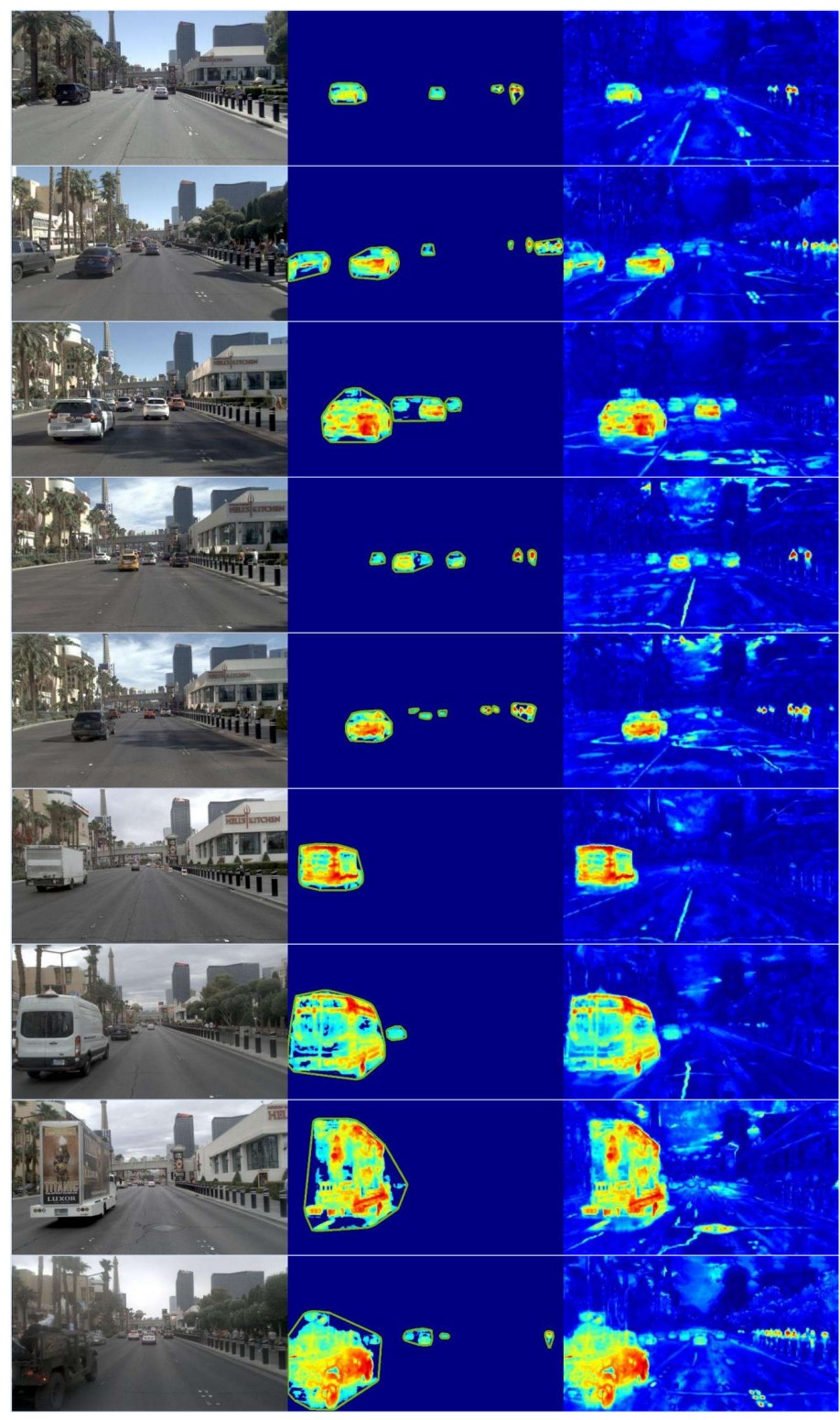

Figure XX: **Qualitative results of** `EmerSeg` **for multiple traversals of location 5 of Mapverse-nuPlan.** From left to right: raw RGB image, extracted 2D ephemeral object masks, and normalized feature residuals visualized using a jet color map.

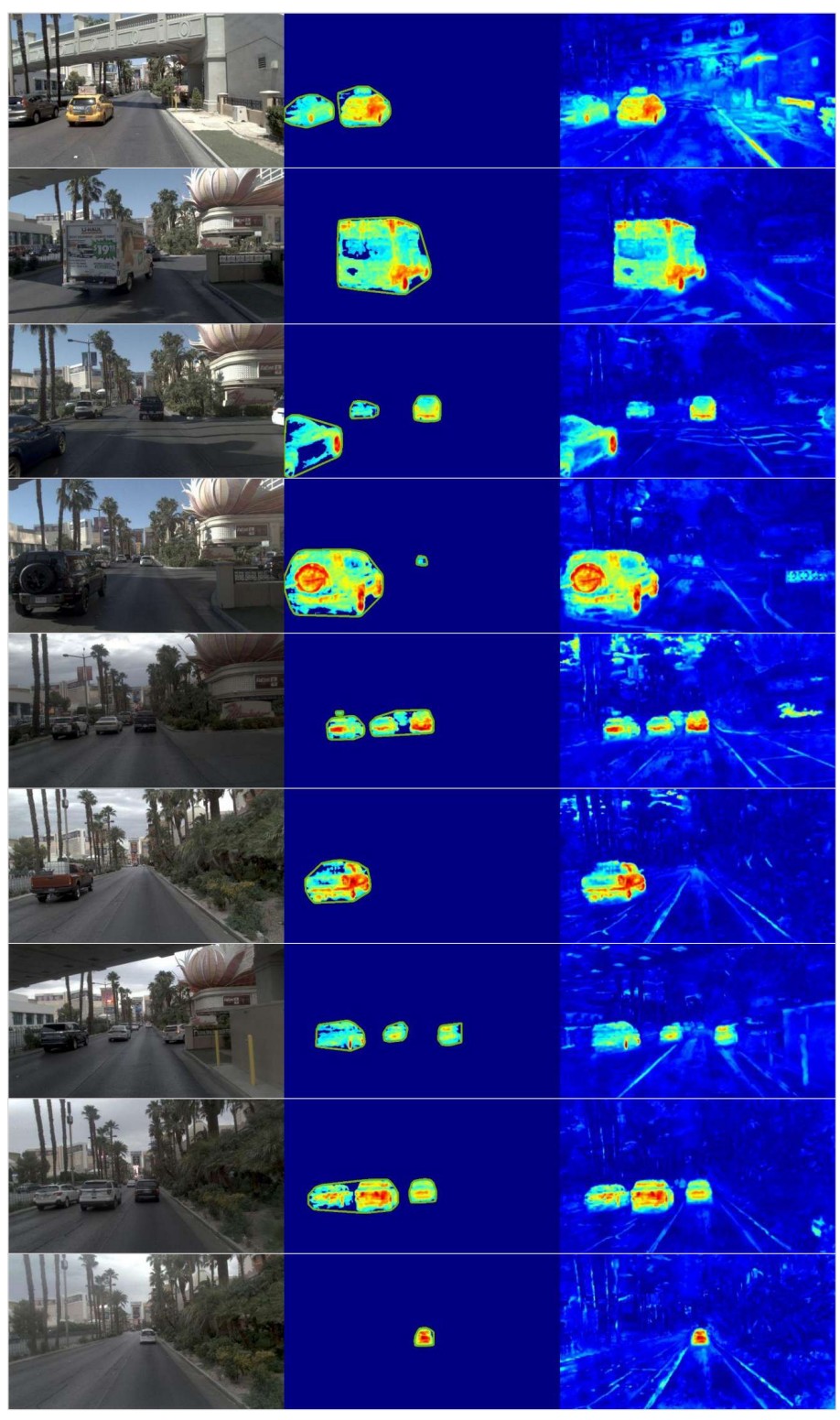

Figure XXI: **Qualitative results of** `EmerSeg` **for multiple traversals of location 6 of Mapverse-nuPlan.** From left to right: raw RGB image, extracted 2D ephemeral object masks, and normalized feature residuals visualized using a jet color map.

# G  Mapverse-nuPlan: Depth Visualization

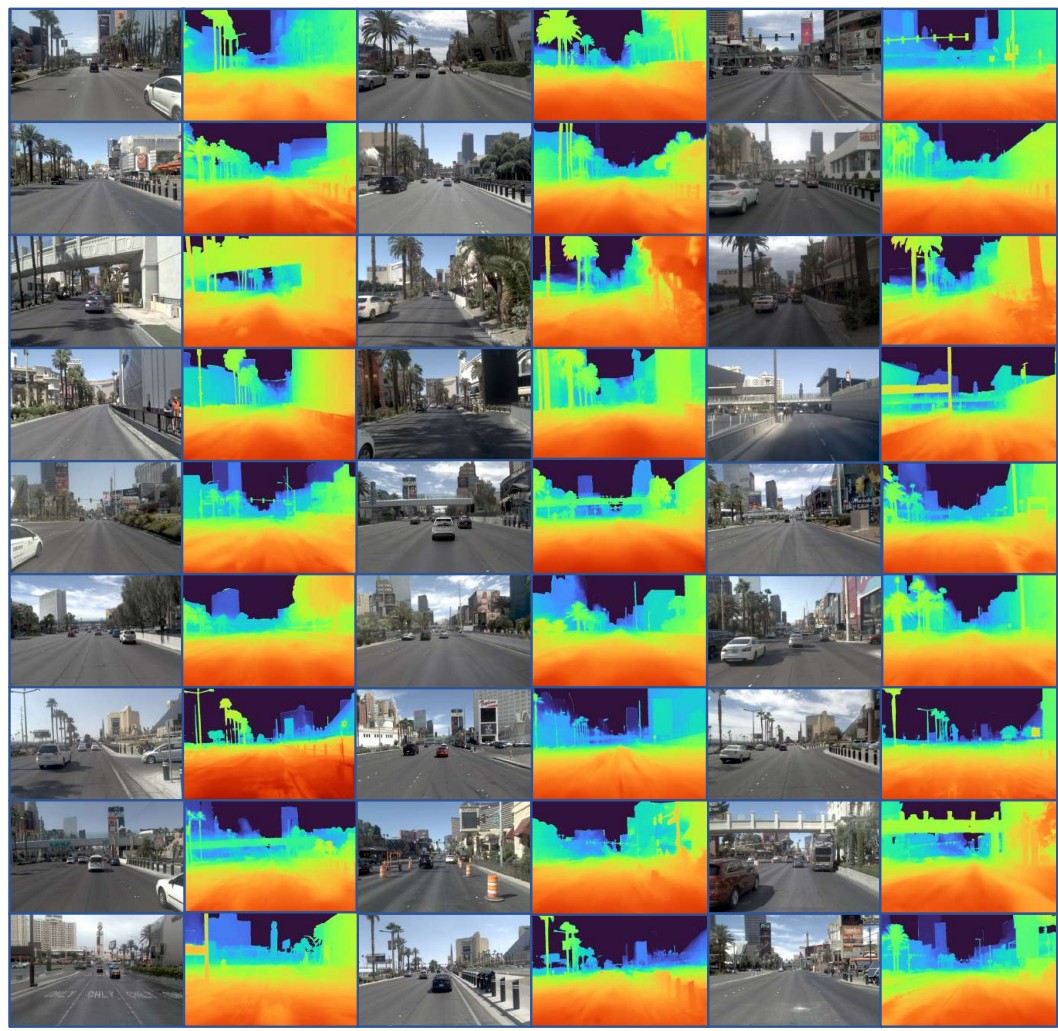

Figure XXII: **Visualizations of depth images in Mapverse-Ithaca365**

# H    Mapverse-nuPlan: Neural Rendering

Table II: **Quantitative rendering results in Mapverse-nuPlan.** We set test/training views as 1/8. Pixels corresponding to transient objects are removed in the evaluations since we do not have ground truth background pixels in these regions occluded by transient objects.

| Location | 3DGS | | | 3DGS+SegFormer | | | EnvGS (Ours) | | |
|---|---|---|---|---|---|---|---|---|---|
| | LPIPS ($\downarrow$) | SSIM ($\uparrow$) | PSNR ($\uparrow$) | LPIPS ($\downarrow$) | SSIM ($\uparrow$) | PSNR ($\uparrow$) | LPIPS ($\downarrow$) | SSIM ($\uparrow$) | PSNR ($\uparrow$) |
| 1 | 0.177 | 0.812 | 21.94 | 0.164 | 0.820 | 21.99 | 0.167 | 0.818 | 21.88 |
| 2 | 0.162 | 0.836 | 21.22 | 0.154 | 0.840 | 21.28 | 0.154 | 0.839 | 21.26 |
| 3 | 0.173 | 0.821 | 20.59 | 0.165 | 0.827 | 20.77 | 0.164 | 0.826 | 20.77 |
| 4 | 0.174 | 0.843 | 20.99 | 0.156 | 0.850 | 21.12 | 0.156 | 0.849 | 21.09 |
| 5 | 0.160 | 0.828 | 19.82 | 0.143 | 0.835 | 20.20 | 0.144 | 0.835 | 20.17 |
| 6 | 0.263 | 0.765 | 19.41 | 0.240 | 0.778 | 19.85 | 0.240 | 0.778 | 19.85 |
| 7 | 0.232 | 0.772 | 18.92 | 0.228 | 0.777 | 18.55 | 0.227 | 0.777 | 18.56 |
| 8 | 0.161 | 0.823 | 21.23 | 0.158 | 0.825 | 21.17 | 0.158 | 0.825 | 21.16 |
| 9 | 0.171 | 0.826 | 20.77 | 0.161 | 0.832 | 20.98 | 0.162 | 0.831 | 20.97 |
| 10 | 0.163 | 0.844 | 21.75 | 0.151 | 0.854 | 22.26 | 0.156 | 0.850 | 21.93 |
| 11 | 0.179 | 0.827 | 20.86 | 0.169 | 0.832 | 21.08 | 0.170 | 0.831 | 21.06 |
| 12 | 0.187 | 0.815 | 19.93 | 0.172 | 0.824 | 20.22 | 0.172 | 0.824 | 20.22 |
| 13 | 0.253 | 0.786 | 19.74 | 0.239 | 0.792 | 19.98 | 0.241 | 0.792 | 19.96 |
| 14 | 0.189 | 0.798 | 19.50 | 0.181 | 0.803 | 19.73 | 0.180 | 0.804 | 19.72 |
| 15 | 0.224 | 0.823 | 20.58 | 0.193 | 0.836 | 20.99 | 0.194 | 0.836 | 20.99 |
| 16 | 0.154 | 0.846 | 21.98 | 0.144 | 0.851 | 21.94 | 0.145 | 0.851 | 21.91 |
| 17 | 0.171 | 0.844 | 21.73 | 0.153 | 0.850 | 21.67 | 0.158 | 0.848 | 21.59 |
| 18 | 0.207 | 0.803 | 19.36 | 0.187 | 0.811 | 19.69 | 0.186 | 0.812 | 19.68 |
| 19 | 0.212 | 0.797 | 19.18 | 0.206 | 0.802 | 18.99 | 0.206 | 0.802 | 19.02 |
| 20 | 0.173 | 0.840 | 19.64 | 0.142 | 0.854 | 19.94 | 0.143 | 0.854 | 19.92 |
| AVERAGE | 0.189 | 0.818 | 20.46 | 0.175 | 0.825 | 20.62 | 0.176 | 0.824 | 20.59 |

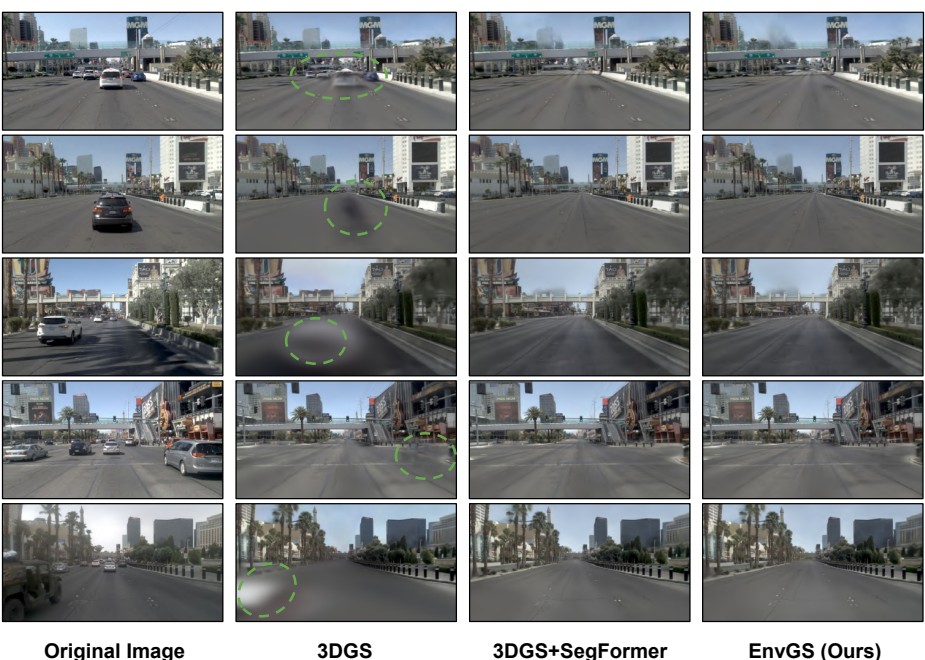

| Original Image | 3DGS | 3DGS+SegFormer | EnvGS (Ours) |

Figure XXIII: **Visualizations of neural rendering in Mapverse-nuPlan.**

# I   Limitations and Future Work

## I.1   Unsupervised 2D Segmentation

**Shadow segmentation**   Figure XXIV illustrates the challenges encountered in accurately segmenting shadows. Each row represents different instances. The left column displays the original images, the middle column presents the segmentation output, and the right column highlights the areas where shadow removal failed, indicated by red circles. While there are some successful cases marked by green circles, our method lacks consistency across different scenes.

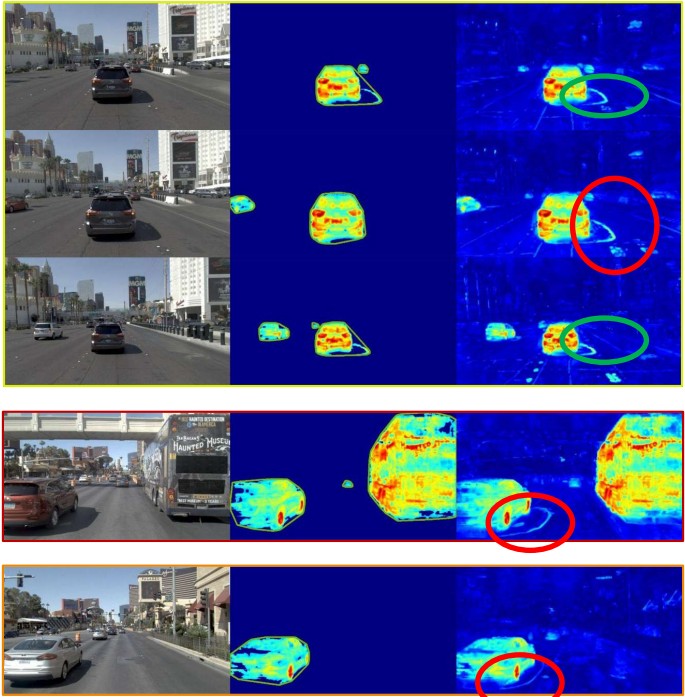

Figure XXIV: **Failure cases of shadow segmentation.**

**Large occluders**   Figure XXV illustrates the challenges faced when segmenting scenes with large and enduring occluders. When occluders occupy a significant portion of pixels and persist over time, our model tends to overfit to these occluders. This leads to a reduction in the feature residuals of the corresponding pixels, thereby failing the segmentation.

**Long-range objects**   Figure XXVI highlights the challenges encountered when segmenting scenes with small and long-range objects. The red circles indicate regions where the segmentation algorithm struggles to differentiate these objects from their surroundings, often missing or inaccurately segmenting them.

**Reflective Surfaces**   Figure XXVII highlights the model's current limitations in handling reflective surfaces, which can vary significantly across traversals due to changes in lighting.

**Future Work**   Future work will focus on developing better methods to robustly segment object shadows and leveraging temporal information to more effectively handle large and enduring occluders. Additionally, designing adaptive thresholds based on object distance will help better exploit the spatial information of the feature residuals. Furthermore, training a vision foundation model using large-scale, in-the-wild data will be crucial for enhancing the model's robustness.

## I.2   Geometry Reconstruction

There are still challenges in road reconstruction, particularly due to the textureless nature of road surfaces. To address this, integrating advanced techniques such as mesh reconstruction [30] and 2D

Gaussian Splatting [85] could significantly enhance the geometric reconstruction capabilities of our method. By enhancing the geometric fidelity of road surfaces, these techniques can help overcome the limitations posed by textureless areas, ensuring a more comprehensive and reliable mapping of driving environments.

## I.3 Neural Rendering

Our method currently struggles with handling significant lighting variations and seasonal changes in the environment. Incorporating a 4D representation [28, 44], which accounts for changes over time, could further enhance the quality of neural rendering. Additionally, we have not yet investigated very large-scale scene reconstruction. Incorporating recent Level-of-Detail (LOD) techniques can help address this large-scale problem [88]. We leave these as future works.

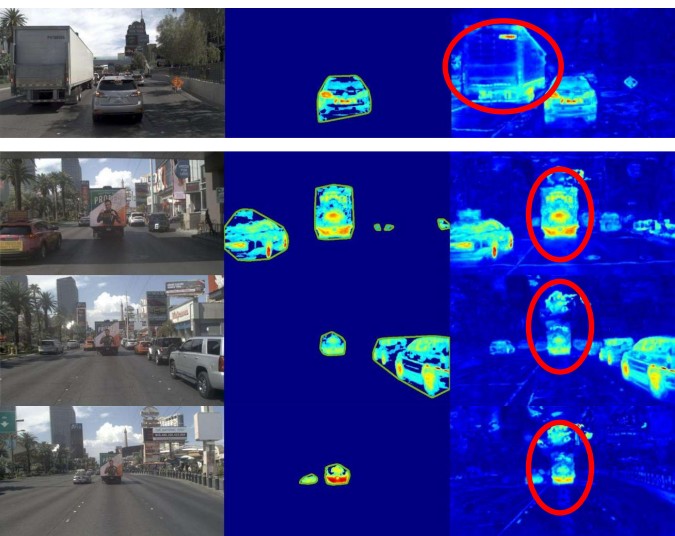

Figure XXV: **Failure cases when faced with large and enduring occluders.**

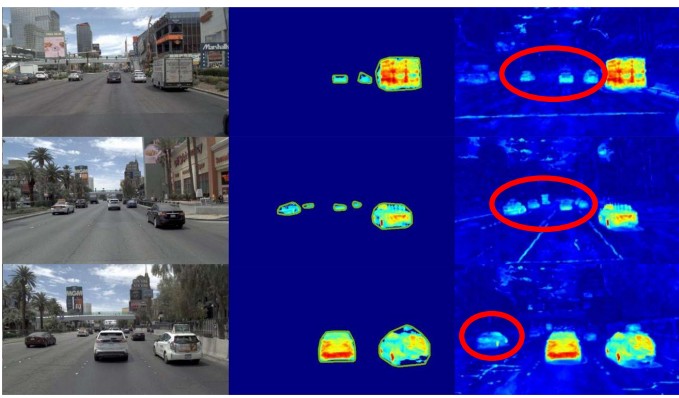

Figure XXVI: **Failure cases when faced with small and long-range objects.**

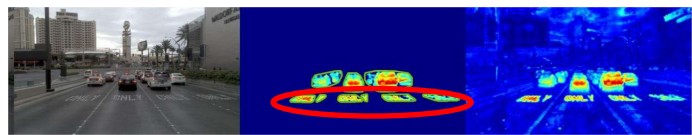

Figure XXVII: **Failure cases when faced with reflective surfaces.**

