# OpenReview forum: "Memorize What Matters: Emergent Scene Decomposition from Multitraverse"
_NeurIPS.cc/2024/Conference — NeurIPS 2024 spotlight_

### Official Review · Reviewer_mMjd · 2024-07-04

**Soundness:** 2
**Presentation:** 3
**Contribution:** 2
**Rating:** 5
**Confidence:** 5

**Summary:**

This paper presents a 3D Gaussian mapping framework that is able to convert multitraverse videos from the same region into a environment while segmenting out 2D ephemeral objects. Leveraging the mlutitraverse data, the scene decomposition emerges in an unsupervised manner, as a result of the consensus in background and dsisensus in transient foreground across traversals. The paper demonstrate promising results in object segmentation, 3D reconstruction, and neural rendering.

**Strengths:**

+ The paper is overall well written and easy to follow.
+ The proposed approach is self-supervised, which does not rely on any object annotations, and hence holds good potential for scalability.
+ The paper demonstrates the value of multitraverse data in scene decomposition, which is relatively under exploited in prior work.

**Weaknesses:**

- The method performs segmentation and finally GS mapping sequentially, which means the error in segmentation would propagate into the subsequent GS mapping. Why not perform multiple rounds of segmentation and GS mapping to continue improving the robustness against outliers?
- The novelty of the paper is somewhat limited. Similar idea of emergent scene decomposition has been demonstrated in prior work EmerNeRF, where  DINO features are also used. This paper differs mostly in using multitraverse data, and adopting 3DGS as scene representation instead of NeRF.
- In object segmentation task, the paper compare with unsupervised segmentation method, but lacks comparisons with object discovery methods. As noted in Sec. 6, the proposed method is highly relevant to object discovery.
- The comparison with STEGO and CAUSE is in a sense unfair because they do not use multitraverse data. A simple baseline would be performing a matching (e.g. with SIFT) for each segmented region with other traversals, a failure of which indicates the region being transient and vice versa.
- For 3D environment reconstruction, only depthanything is compaired, which is however a single-image depth estimation, whereas the proposed method is multiview based. How about comparing with SOTA multiview stereo approaches?  In addition, why is Chamfer Distance adopted as evaluation metric? Chamfer distance is a weaker metric as it does not leverage correspondence to the ground truth, but here we do have pixel-level correspondence by projecting Lidar to the image plane. Lastly, is the depth evaluation carried out only on background regions? There is no explanation on this regard.
- As shown in Table 3, the neural rendering performance of EnvGS is only marginally better than the original 3DGS. It is even lower than 3DGS in PSNR.

**Questions:**

The questions to be answered are detailed in the weakness section, mainly on stronger baselines in object segmentation, and evaluation on 3D mapping.

**Limitations:**

The paper has discussed the limitations.

---

> ### Author Rebuttal · Authors · 2024-08-06
>
> We sincerely appreciate your constructive feedback and helpful suggestions. Please find our detailed response below.
>
> ---
> *Q1. Multiple rounds*
>
> We have added one additional round of segmentation and mapping by incorporating the emerged masks into COLMAP to remove transient objects during Gaussian initialization. This resulted in improved rendering scores, with PSNR increasing from 23.11 to 23.26 and SSIM from 0.8299 to 0.8372 in 1000 images at one location. We will include more experiments using multiple rounds. We appreciate your insightful comments.
>
> ---
> *Q2. EmerNeRF*
>
> We would like to emphasize that our method is fundamentally different from EmerNeRF, not just in using multitraversal data and 3DGS. Although both our method and EmerNeRF address self-supervised scene decomposition, the key ideas are entirely different: **EmerNeRF learns static-dynamic separation by effective parameterization of corresponding fields and leverages scene flow as an inductive bias, whereas our method leverages multitraversal consensus as a self-supervised signal to decompose transient objects from the permanent environment.** As a result, our method can decompose not only dynamic objects but also static yet movable objects, e.g., parked cars. In contrast, EmerNeRF can only decompose moving objects. Besides, **our method only needs camera input while EmerNeRF requires LiDAR. Our insight is to leverage more camera observations to boost the reconstruction performance.**  In summary, compared to EmerNeRF, our method has three differences:
>
> * Our method introduces the **conceptually novel idea of multitraversal consensus-based scene decomposition**.
> * Our method  provides a **more powerful scene decomposition** approach, capable of decomposing both moving and static yet movable objects.
> * Our method  adopts a more **cost-effective** sensor setup.
>
> We have added quantitative and qualitative results of EmerNeRF. We found that EmerNeRF cannot remove static cars because it requires motion to decompose dynamic parts. Additionally, the decomposition performance is not ideal, likely due to the monocular camera input. The segmentation IoU is **7.3%**, which is significantly lower than our method (>40%). Visualizations are shown in Figures 6 and 7 in the PDF under the global rebuttal.
>
> We hope our clarifications address your concerns regarding the novelty of our work. *As agreed upon by the other two reviewers, our method is an innovative approach with key insights.*
>
> ---
> *Q3. Object discovery*
>
> We agree that adding comparisons to object discovery methods would make our experiments more comprehensive. We have included results from FOUND [1] which is a SOTA unsupervised object discovery method and found them to be much worse than our method, with an IoU of only **12.77%**. Some qualitative examples are shown in Figure 8 in the PDF under the global rebuttal. **Another notable advantage of our method compared to object discovery methods is that it not only discovers objects in 2D but also obtains a 3D representation of the static environment.** We will add more results and an expanded literature survey.
>
> [1] Unsupervised object localization: Observing the background to discover objects. CVPR 2023.
>
> ---
> *Q4. Segmentation baselines*
>
> We agree that adding more baselines using multiple traversals would make our experiments more comprehensive. We have added another baseline method: we directly matched DINO features across traversals and identified patches with few matching counterparts in other traversals as transient. The results were less convicing, with an IoU of **10.17%**. Visualizations are shown in Figure 9 in the PDF under the global rebuttal.
>
> We would like to emphasize that the key difference between our method and existing methods is that **we lift 2D to 3D, which facilitates the identification of 2D objects through 3D representation learning.** As a result, *our method not only segments objects in 2D with high quality but also obtains a 3D environment map.* We will include more baseline results and discussions.
>
> ---
> *Q5. Multiview stereo and depth metrics*
>
> Our method is a 3DGS-based approach that integrates both geometric and photometric information. It can be used not only for depth estimation but also for various tasks such as rendering and segmentation. Therefore, **our method is more versatile compared to monocular or multiview depth estimation methods.**
>
> Additionally, our method is a camera-only solution, whereas learning-based depth estimation methods require LiDAR-supervised training. Hence, **our method requires a more cost-effective sensor setup compared to depth estimation methods.**
>
> We have also added a SOTA multiview stereo baseline, DUSt3R [1]. We agree that including depth metrics makes the evaluations more comprehensive. The mean depth RMSE of our method is 7.376m, while DepthAnything is 13.562m, and DUSt3R is 13.37m (background regions). Some qualitative examples of DUSt3R are shown in Figure 10 in the PDF under the global rebuttal. These results demonstrate that our method remains the best without the need for LiDAR sensors. We will add the mutliview stereo results and depth metrics in the final version of our paper. **Note that our method and depth estimation can be complementary: our method can provide pseudo ground truth for depth estimation, while depth estimation can aid in the initialization stage of our method. Future work includes leveraging multitraversal to enhance depth estimation methods and utilizing depth estimation for efficient 3DGS initialization.**
>
> [1] DUSt3R: Geometric 3D Vision Made Easy. CVPR 2024
>
> ---
> *Q6. Table 3*
>
> Table 3 does not evaluate pixels corresponding to transient objects, as we do not have ground truth background pixels in regions occluded by transient objects. Therefore, the observed gap is not too large. However, our method performs significantly better in occluded regions, as shown in Figures 6 and XIV.

---

### Official Review · Reviewer_3Lqq · 2024-07-12

**Soundness:** 3
**Presentation:** 3
**Contribution:** 2
**Rating:** 5
**Confidence:** 2

**Summary:**

The paper presents a novel approach for self-supervised scene decomposition using multi-traverse camera data, which results in a high-quality static background scene reconstruction via Gaussian Splatting. The method 3D Gaussian Mapping leverages repeated traversals and feature distillation to capture the emergent focus on 2D consensus structures and, therefore, dynamic foreground masks, which contribute to mitigating the disturbance of temporal dynamic objects on visual-based 3D mapping. Along with the proposed benchmark, this method paves an interesting direction, leveraging traversals to learn the inherent structure of background in autonomous driving, which finds wide applications in 3D map construction, map change detection, driving simulation, etc.

**Strengths:**

1. **Innovative Approach**: The combination of multitraversal and Gaussian Splatting is a fresh and compelling method. It effectively decomposes an urban scene into static and dynamic elements merely via visual features and image rendering, which is an innovative idea for the urban scene decomposition task.

2. **Method of Simplicity and Effectiveness**: The paper combines and adapts the latest feature distillation, denoised DINOv2, and 3D reconstruction methods, Gaussian Splatting, to accomplish self-supervised dynamic component segmentation through emergent outweighing effect by multitraversal data. Visual feature residuals resulting from rendering results are effectively utilized to extract contours of the ephemeral elements simply by spatial gradient with the following postprocessing.

3. **Comprehensive Evaluation and Ablation Study**: The evaluation section is thorough, demonstrating the method's effectiveness across three tasks, i.e., 2D segmentation, 3D reconstruction, and 3D mapping. The abundant qualitative and quantitative results show robust performance under diverse conditions and improvements over existing techniques, particularly in ephemerality decomposition and 3D scene geometry learning. An encompassing ablation study shows the influence of traversals and selection of hyperparameters on segmentation tasks.

**Weaknesses:**

1. **Assumptions on Environmental Stability**: The method strictly assumes a stable environment without major geometry change under consistent illumination and weather. This assumption might not hold in wider real-world scenarios, potentially affecting the method's robustness and generalizability.

2. **Lack of Failure Case Study**: Even though the paper discusses some failure cases regarding shadow, occlusion, reflection, etc. It's still interesting to see the influence of weather and large illumination changes on this approach and if increasing the number of traversals can tackle the issue.

3. **Lack of Comparative Baselines Regarding Env Reconstruction and Rendering**: Although the paper provides a solid evaluation of all tasks, it could benefit from more comparative analysis with state-of-the-art urban scene reconstruction methods, especially regarding static environment representation.

**Questions:**

1. **Additional Comparison to Urban Scene Reconstruction Methods**: Some comparisons to the latest urban scene reconstruction methods, like EmerNeRF, HUGS, NeuRad, and so on, regarding the static env reconstruction and rendering quality, especially to the results of actor removal. This can serve as a broader discussion on the effect of multitraversals and sensor types.

2. **Failure Case Study**: Despite the strict assumption of this approach, it's still compelling to discuss the effect of environment and illumination change on this approach.

3. **More Details on Training and Rendering Procedure** This paper offers a detailed workflow as an overview of the whole approach. However, directly embedding Dino features in 3DGS can have side effects on training and rendering, especially on this large scene. More details on hyperparameter selection and the training process can be helpful for further work in this field.

Overall, this paper presents a promising and innovative advancement in scene decomposition from multitraverse data, with several strengths that make it a valuable contribution to the field of computer vision. Addressing the noted weaknesses could further enhance its impact and generality.

---

> ### Author Rebuttal · Authors · 2024-08-06
>
> We appreciate your valuable feedback and insightful suggestions. Below is our detailed response.
>
> ---
> *Q1: Assumptions on Environmental Stability: The method strictly assumes a stable environment without major geometry change under consistent illumination and weather.*
>
> We agree with you that relaxing the assumption of environmental stability can further enhance the impact and generality of our method. We have made two efforts to address your concern:
>
> * We have tested our method under more **challenging illumination and weather conditions, such as night, foggy, and rainy** scenarios. Specifically, we added images from one additional traversal under adverse conditions to the multitraversal collection. Surprisingly, our method still performed well in unsupervised segmentation, even under night, foggy, and rainy conditions, as shown in Figure 1-3 in the PDF under the global rebuttal. **This demonstrates that our method is robust against inconsistent illumination and weather conditions, including night, fog, and rain.**
> * We have incorporated a learnable traversal embedding into the 3DGS. This effectively models the appearance changes across traversals. Due to the limited time for the rebuttal, we present preliminary results in Figure 4 in the PDF under the global rebuttal. From the top left to the bottom right, it shows the rendering results by interpolation between two traversal embeddings.
>
> We will include these experiments in the final version of our paper. Additionally, we would like to emphasize that **3DGS for self-driving with multiple traversals is an unexplored research topic, especially considering significant environmental changes (adverse weather). There are many open questions in this domain, and we believe our dataset and method can serve as a benchmark and baseline to inspire further research on this topic.**
>
> ---
>
> *Q2: Lack of Failure Case Study: Even though the paper discusses some failure cases regarding shadow, occlusion, reflection, etc. It's still interesting to see the influence of weather and large illumination changes on this approach and if increasing the number of traversals can tackle the issue.*
>
> We agree with you that discussing the influence of weather and large illumination changes would make our experiments more comprehensive. We have conducted additional experiments with traversals collected under challenging weather and illumination conditions. As discussed above, we find that **our method demonstrates good robustness against diverse conditions**. Note that these experiments use 11 traversals, and decreasing the number of traversals will degrade segmentation performance, similar to the findings in Ablation Study C.4 in our paper (Ablation Study on Number of Traversals: Visualization and Discussion).
>
> In addition, we have also added one traversal with snow on the road. We observed that snow on the road can also be segmented, as it appears only in a single traversal, as shown in Figure 5 in the PDF under the global rebuttal. We do not consider this a "failure case" since snow is transient. We will include these results in the final version of our paper.
>
> ---
>
> *Q3: Lack of Comparative Baselines Regarding Env Reconstruction and Rendering: Although the paper provides a solid evaluation of all tasks, it could benefit from more comparative analysis with state-of-the-art urban scene reconstruction methods.*
>
> We agree that adding additional comparisons to urban scene reconstruction methods would enhance our experiments. There are several fundamental differences between our method and existing ones such as EmerNeRF, HUGS, and NeuRad. We will include these comparisons in the final version of our paper to provide a broader discussion on the effect of multitraversals and sensor types.
>
> * **Input and Output**: These works take a single-traversal video as input, which limits their ability to reconstruct static environments occluded by static yet transient objects, such as parked cars. *In contrast, our method, based on multi-traversal input, can reconstruct a static environment without any movable objects (including both dynamic and static vehicles).*
>
> * **Sensor Requirements**: These methods require LiDAR point cloud data as input. In contrast, our method only uses RGB images, using a more *cost-effective and portable* sensor setup.
>
> * **Supervision**: HUGS and NeuRad require 3D bounding boxes as input, making them not fully self-supervised methods. *Our method, however, does not rely on such external supervision.*
>
> We have added quantitative and qualitative results of EmerNeRF, a self-supervised method, for a fair comparison. We ran the original codebase of EmerNeRF with both LiDAR and camera inputs on each traversal in our dataset.
>
> * **Actor Removal**: EmerNeRF cannot remove static cars because it models static cars as backgrounds and only segments objects with motion. Besides, the decomposition performance is not ideal, likely due to the monocular camera input. The segmentation IoU is **7.3%**, which is significantly lower than our method.
> * **Adverse Weather**: The performance under adverse weather conditions is unsatisfactory, as shown in the rainy day example in Figure 4 in the PDF under the “global” response.
> * **Rendering Quality**: The rendering quality is also inferior to our method, as shown in Figure 6 and Figure 7 in the PDF under the global rebuttal. We will include more quantitative results in the final version of our paper.
>
> ---
>
> *Q4: More Details on Training and Rendering Procedure.*
>
> We agree that providing more details can be valuable for further work in this field. Most of our method follows the hyperparameters of 3DGS. Empirically, we found that the feature rendering loss function plays a crucial role in scene decomposition: the KL divergence loss function performs much better than the L1 loss. We will report more ablation studies in the final version of our paper and release our code to support further research.

---

> > ### Comment · Reviewer_3Lqq · 2024-08-13
> > **Thanks for the response**
> >
> > Hi, I appreciate the detailed responses to my concerns. Thanks for agreeing with me in many aspects. I am happy to stick on my initial rate.

---

> > > ### Author Response · Authors · 2024-08-13
> > > **Thank you for recognizing our rebuttal and contributions to the field.**
> > >
> > > Dear Reviewer 3Lqq,
> > >
> > > Thank you for your follow-up comment and for acknowledging our detailed responses to your concerns. We are glad that our additional experiments and explanations addressed your concerns. We also appreciate your recognition of our contributions to the field. We respect your decision to maintain your initial rating and are grateful for your thoughtful review and feedback, which has helped us improve our paper.
> > >
> > > Thank you again for your time and effort in reviewing our work!
> > >
> > > Best regards,
> > >
> > > NeurIPS 2024 Conference Submission3651 Authors

---

### Official Review · Reviewer_D2oY · 2024-07-13

**Soundness:** 4
**Presentation:** 3
**Contribution:** 3
**Rating:** 7
**Confidence:** 5

**Summary:**

The paper proposes a method called 3DGM that performs foreground-background disentangled 3D reconstruction by capturing the consistent parts from multi-traverse videos. 3DGM leverages 3DGS as the scene reconstruction algorithm, using only camera images as input, and achieves decoupled reconstruction of the 3D environment and 2D object segmentation. Additionally, the paper introduces a new dataset that combines Ithaca365 and nuPlan, which is used to evaluate unsupervised 2D segmentation, 3D reconstruction, and neural rendering.

To be specific, the author has observed that self-driving cars often traverse the same routes repeatedly, encountering new pedestrians and vehicles each time, similar to how humans encounter different people every day in the same 3D environment. Inspired by the fact that humans are better at remembering permanent structures and forgetting transient objects, the author proposes the idea of developing a mapping system that can identify and memorize the consistent structures of the 3D world through multiple traversals without the need for human supervision.

The key insight is that while the specific pedestrians and vehicles change from one traversal to the next, the underlying 3D environment (buildings, roads, etc.) remains the same. By leveraging this observation, the proposed system could learn to filter out the transient objects and focus on mapping the permanent structures, akin to how humans naturally encode spatial knowledge. The goal is to create a self-supervised system that can build a robust 3D map of the environment simply by repeatedly navigating through it without requiring any manual labeling or annotations.

**Strengths:**

3DGM is an unsupervised approach that doesn't require any additional manual annotations. The reconstruction process relies solely on camera input, without the need for depth measurements from sensors like LiDAR. Employing 3DGS enables much faster reconstruction speeds for implicit neural rendering. The authors also contribute a new multi-traverse video dataset.

**Weaknesses:**

The end-to-end process for 3DGM appears to be quite complex, with multiple stages involved. However, the paper fails to provide a lucid explanation of the nitty-gritty details involved in implementing the algorithms. Without the availability of the accompanying source code, it would be an uphill task for others to replicate the results or build upon this work effectively.

**Questions:**

Please check the Weaknesses and Limitations

**Limitations:**

Firstly, although this is a self-supervised approach, in practical applications, multiple passes along the same road are still required for reconstruction. Could some prior information be incorporated to enable the model to infer the background and complete the task with fewer traversals?

Secondly, the methods based on 3D reconstruction have high data quality requirements, and the 3DGS-based methods need a reasonably accurate initial pose estimation (by COLMAP), which further exacerbates the demand for high-quality data.

---

> ### Author Rebuttal · Authors · 2024-08-06
>
> We are grateful for your insightful feedback and suggestions. Please review our detailed response below.
>
> ---
> *Q1. The end-to-end process for 3DGM appears to be quite complex, with multiple stages involved. However, the paper fails to provide a lucid explanation of the nitty-gritty details involved in implementing the algorithms. Without the availability of the accompanying source code, it would be an uphill task for others to replicate the results or build upon this work effectively.*
>
> We would like to emphasize that, although our method has multiple stages, each module is **easy to deploy** and **computationally efficient**. As discussed in the paper, using a feature map with a resolution of 110×180 requires only 2,000 iterations to achieve an IoU score exceeding 40%, taking approximately 8 minutes on a single NVIDIA RTX 3090 GPU for 1,000 images from 10 traversals of a location. Future work will include investigating more efficient initialization methods. A detailed workflow of the overall method is provided in Appendix A. To support reproducibility and further research, we will release our source code and dataset upon acceptance of our work.
>
> ---
>
> *Q2. Firstly, although this is a self-supervised approach, in practical applications, multiple passes along the same road are still required for reconstruction. Could some prior information be incorporated to enable the model to infer the background and complete the task with fewer traversals?*
>
> We agree that incorporating prior information can further enhance the efficiency of the proposed method. For example, if past traversals with both LiDAR and camera data are available, the 3D reconstruction could benefit significantly from such prior information. **In fact, our method could serve as a strong camera-only baseline for constructing such a scene prior, facilitating scene decomposition and reconstruction in future traversals of the same location.** *We leave the exploration of prior information for future work, as it requires a complete reformulation of the problem, which is non-trivial and beyond the scope of this work.* **Future research opportunities include developing algorithms that seamlessly integrate prior data, exploring the impact of different types of prior information on reconstruction quality, and creating frameworks that can dynamically adapt to changes in new traversals.** We will include more discussions on this in the final version of our paper.
>
> In addition, we would like to emphasize that the setup of multiple passes of the same location is quite feasible, as vehicles typically operate within the same spatial region. As shown in a recent publication at CVPR 2024 [1], it is possible to obtain hundreds of traversal data for the same location within several days.
>
> ---
>
> *Q3. Secondly, the methods based on 3D reconstruction have high data quality requirements, and the 3DGS-based methods need a reasonably accurate initial pose estimation (by COLMAP), which further exacerbates the demand for high-quality data.*
>
> Our method only requires **monocular** RGB images with a resolution of around **900x600**, which can be **easily obtained and is the most cost-effective data collection method**. *No other sensor data, such as LiDAR point clouds, GPS, or IMU data, are needed.* Additionally, **data collection by multiple traversals of the same location is highly feasible, as demonstrated by existing datasets collected by either academic labs or industry companies, such as Ithaca365, nuPlan, and MARS [1].**
>
> In addition, we have found that multitraversal RGB images can facilitate COLMAP initialization, producing very accurate camera poses. Based on our empirical studies, **COLMAP initialization requires only 2 or 3 traversals with only monocular RGB images** to significantly improve the success rate compared to single-traversal scenarios.
>
> &nbsp;
>
> [1] Li, Y., Li, Z., Chen, N., Gong, M., Lyu, Z., Wang, Z., Jiang, P. and Feng, C., 2024. Multiagent Multitraversal Multimodal Self-Driving: Open MARS Dataset. In Proceedings of the IEEE/CVF Conference on Computer Vision and Pattern Recognition (pp. 22041-22051).

---

### Author Rebuttal · Authors · 2024-08-06

## Global Rebuttal
We sincerely thank all the reviewers for their insightful comments. We appreciate the positive feedback: **fresh and compelling, innovative approach, evaluation section is thorough, promising and innovative advancement in scene decomposition, valuable contribution to the field of computer vision, well-written and easy to follow, and good potential for scalability.** Major concerns raised by our reviewers are (1) data requirement, (2) robustness against lighting and illumination changes, (3) comparison to HUGS and NeuRad, (4) comparison to EmerNeRF, (5) comparison to unsupervised object discovery and segmentation, and (6) comparison to depth methods. We summarize our responses below to fully address these concerns.

---
### 1. Data Requirement
Our method only requires **unposed monocular RGB images** with a resolution of **900x600**, which can be easily obtained and is the **most cost-effective** data collection method. *No other data, such as LiDAR point clouds, GPS, and IMU data, are needed*. Additionally, data collection by multiple traversals of the same location is highly feasible, as demonstrated by existing datasets collected by either academia or industry, such as Ithaca365 [CVPR'22] by Cornell University, nuPlan [ICRA'24] by Motional, and MARS [CVPR'24] by May Mobility.

---
### 2. Robustness Against Lighting and Illumination Changes
We have tested our method under challenging conditions such as **night, fog, snow, and rain**. **The results demonstrated our method's robustness against diverse illumination and weather conditions** (see Figure 1-3). We also incorporated a learnable traversal embedding to model appearance changes across traversals. Preliminary results show that this approach effectively handles appearance changes (see Figure 4), and we will include these results in the final paper.

---
### 3. Comparison to HUGS and NeuRad

Our method differs from HUGS and NeuRad in several key aspects:

- **Input and Output:** HUGS and NeuRad require single-traversal videos, which limits their ability to reconstruct static environments **occluded by static yet transient objects**. Our multitraversal input approach overcomes this limitation.
- **Sensor Requirements:** HUGS and NeuRad **need LiDAR point cloud data**, whereas our method only requires RGB images.
- **Supervision:** HUGS and NeuRad **require 3D bounding boxes**, making them not fully self-supervised. Our method does not rely on external supervision.

We believe the above fundamental differences can already distinguish our method from these supervised and LiDAR-based methods. We will add more discussions.

---
### 4. Comparison to EmerNeRF

EmerNeRF and our method both address self-supervised scene decomposition, but they differ in three aspects:

- **Key Idea:** EmerNeRF uses **scene flow** as a self-supervised signal to separate dynamic objects from static backgrounds, while we leverage **multitraversal consensus** as a self-supervised signal for transient-permanent decomposition.
- **Sensor:** EmerNeRF requires **LiDARs**, while we **only use RGB images**, providing a more cost-effective and portable solution.
- **Functionality:** EmerNeRF can only decompose **moving objects**, whereas our method can handle **both dynamic and static yet movable objects**, e.g., parked vehicles.

We demonstrate that our method outperforms EmerNeRF, which requires both camera and LiDAR input. Our approach achieves a segmentation IoU of over 40%, compared to EmerNeRF's 7.3%. While EmerNeRF struggles with static cars and produces noisy decompositions (see Figure 6-7), our method maintains high performance even in adverse weather conditions.

---
### 5. Comparison to Unsupervised Object Discovery and Segmentation

We compared our method to FOUND [CVPR'23], a state-of-the-art unsupervised object discovery method, which showed inferior performance with an IoU of 12.77% (see Figure 8). We also implemented a baseline using DINO-based feature matching across traversals. However, this approach resulted in poor and noisy segmentations, with an IoU of 10.17% (see Figure 9). **Note that our method not only discovers/segments objects in 2D with high quality (with an IoU of >40%) but also reconstructs the 3D environment, offering a significant advantage.** **Our key novelty is to lift 2D to 3D so that we can achieve more accurate 2D segmentation by learning representations in 3D.**

---
### 6. Comparison to Monocular and Multiview Depth Estimation Methods

We would like to emphasize that our method is a 3DGS-based approach that can be used not only for depth estimation but also for various tasks such as rendering and segmentation. Therefore, **our method is more versatile compared to monocular or multiview depth estimation methods**. Additionally, our method is a camera-only solution, whereas learning-based depth estimation methods require LiDAR-supervised training. Hence, **our method requires a more cost-effective sensor setup compared to depth estimation methods.**

*Note that our method and depth estimation can be complementary to each other: our method can provide pseudo ground truth for depth estimation, while depth estimation can facilitate the initialization stage of our method.*

In addition, we have compared our method to DUSt3R [CVPR'24], a state-of-the-art multiview stereo method  (see Figure 10). Our method achieved a mean depth RMSE of 7.376m, significantly better than DepthAnything (13.562m) and DUSt3R (13.37m). These results will be included in the final version of our paper.

---
### Conclusion

In summary, we are the first to achieve **simultaneous 2D segmentation and 3D mapping with pure self-supervision using camera-only input**. *Previous works typically addressed these problems separately and required external supervision or LiDAR sensors.* We believe our method and dataset will inspire future research and significantly contribute to the field. We will release all resources to the community upon acceptance.

---

### Comment · Area_Chair_xXX2 · 2024-08-12
**To reviewers**

Dear reviewers,

This paper received mixed ratings, the author(s) have provided a rebuttal. It would be appreciated if reviewers could carefully read the rebuttal and let the authors and the AC know if it has changed your mind (and detailed justification). Your opinion will play a significant role in the final decision of this paper. Thank you so much.

Best Regards,

AC

---

### Decision · Program_Chairs · 2024-09-25

**Decision:**

Accept (spotlight)

**Comment:**

This paper received a unanimous acceptance recommendation from reviewers after rebuttal.
The AC has checked the comments and agreed with the reviewers, hence an obvious acceptance.